# TOPOLOGICAL DATA ANALYSIS ON NOISY QUANTUM COMPUTERS

**Ismail Yunus Akhalwaya**[†][*]  **Shashanka Ubaru**[†][*]  **Kenneth L. Clarkson**[†]

**Mark S. Squillante**[†]  **Vishnu Jejjala**[‡]  **Yang-Hui He**[§]  **Kugendran Naidoo**[‡]

**Vasileios Kalantzis**[†]  **Lior Horesh**[†]

## ABSTRACT

Topological data analysis (TDA) is a powerful technique for extracting complex and valuable shape-related summaries of high-dimensional data. However, the computational demands of classical algorithms for computing TDA are exorbitant, and quickly become impractical for high-order characteristics. Quantum computers offer the potential of achieving significant speedup for certain computational problems. Indeed, TDA has been purported to be one such problem, yet, quantum computing algorithms proposed for the problem, such as the original Quantum TDA (QTDA) formulation by Lloyd, Garnerone and Zanardi, require currently unavailable fault-tolerance. In this study, we present NISQ-TDA, a *fully implemented end-to-end* quantum machine learning algorithm needing only a short circuit-depth, that is applicable to high-dimensional classical data, and with provable asymptotic speedup for certain classes of problems. The algorithm neither suffers from the data-loading problem nor does it need to store the input data on the quantum computer explicitly. The algorithm was successfully executed on quantum computing devices, as well as on noisy quantum simulators, applied to small datasets. Preliminary empirical results suggest that the algorithm is robust to noise.

## 1   INTRODUCTION

With the advent of modern technology, the collection of information-rich, high-dimensional data has become prevalent. These high-dimensional datasets are typically characterized by multidimensional correlation structures that are difficult to uncover. Extracting and analyzing such structural information is crucial in machine learning as well as in accelerating scientific discovery. Topological data analysis (TDA) is a powerful unsupervised machine learning technique for the extraction of valuable shape-related features of large datasets (Zomorodian & Carlsson, 2005; Ghrist, 2008; Wasserman, 2018). It represents one of the few data analysis algorithms that can process high-dimensional datasets and reduce them to a small set of local and global signature values that are interpretable and laden with predictive and analytical value. TDA has been shown to be useful in various scientific applications, including machine learning and artificial intelligence (AI) for the analysis of deep neural network architectures (e.g., estimate the capacity (Guss & Salakhutdinov, 2018) and topological complexity (Naitzat et al., 2020) of neural networks); neuroscience (Giusti et al., 2015), where topology is used to reveal intrinsic geometric structures in neural correlations; cosmology (Cole & Shiu, 2018b), where TDA is used for detecting non-Gaussianity of the cosmic microwave background (CMB); and genetics (Rabadán et al., 2020; Mandal et al., 2020b), for predicting phenotypes from gene co-expression or raw genomics data. Despite such progress in some applications, the true potential of TDA has been severely limited because classical algorithms for

---

[*]These authors contributed equally to this work. Corresponding authors
[†]IBM Research, USA and South Africa
[‡]University of the Witwatersrand, South Africa
[§]Royal Institution, UK

TDA have proven to be computationally prohibitive, only mitigated to some extent by sampling or by limiting calculations to low-dimensional properties.

Quantum computers represent one potential approach to address these prohibitive computational requirements of TDA. The power of quantum computers lies in their ability to perform computations in large computational (Hilbert) spaces, accessed via relatively small physical systems (Deutsch, 1985; Lloyd, 1996). With the recognition of this novel computational power in the 1980s (Feynman, 1982), there has been an arduous search for algorithms that achieve significant computational speedups over classical algorithms (Shor, 1994; Grover, 1996; Nielsen & Chuang, 2010). Such quantum algorithms offer the potential to solve problems that can not be solved using conventional computers. Quantum computers outperforming current classical supercomputers has been termed *quantum advantage* in the literature (Bravyi et al., 2018; Arute et al., 2019; Deshpande et al., 2022; Rinott et al., 2022). However, this has not yet been achieved for any problem of practical value.

In a seminal paper, Lloyd et al. (2016) proposed Quantum TDA (QTDA), an algorithm that achieves an expected exponential speedup in solving an approximation of TDA. Recent works (Gyurik et al., 2020; Cade & Crichigno, 2021; Crichigno & Kohler, 2022; Schmidhuber & Lloyd, 2022) have studied the hardness of the approximation problem solved by QTDA, and discussed the conditions under which the algorithm provably enjoys speedup over classical algorithms. Furthermore, this speedup is not overshadowed by the data-loading cost (Aaronson, 2015), which plagues several other quantum algorithms (Harrow et al., 2009; Gilyén et al., 2019), especially those related to machine learning (Biamonte et al., 2017; Schuld et al., 2015). However, the QTDA algorithm still requires long-lasting quantum coherence and low computational error to store and process the loaded data. Indeed, it requires *fault-tolerant* quantum computing (Shor, 1996; Aaronson, 2015; Preskill, 2018), an error-corrected quantum computer needing a very large overhead in resources (number of low-noise qubits and operations) (Arute et al., 2019; Zhao et al., 2020). Many components of the Lloyd et al. (2016) algorithm require fault-tolerance: Grover's search (Grover, 1996), Quantum Phase Estimation (Nielsen & Chuang, 2010), and repeated access to the input data. However, fault-tolerance has not yet been achieved on currently available quantum devices, and is likely several years away from full realization (goo, 2023). Intriguingly, the qubit numbers and noise levels that are currently realized in hardware are not classically simulatable, which raises the question of whether some algorithm could make use of these non-fault-tolerant noisy devices (Noisy Intermediate-Scale Quantum (NISQ) (Preskill, 2018)) for quantum advantage?

In this paper we present a quantum algorithm for solving the same problem as QTDA with an improved runtime, shorter circuit depth, and without fault tolerance requirements. Our NISQ-TDA algorithm solves the principal problem of TDA, estimating the Betti numbers of the given data (Ghrist, 2008). The algorithm only requires pairwise distances of the $n$ data-points as input and outputs an estimate for the (normalized) Betti numbers of the data, which are signature values that describe the shape of the data. However, the calculation of these Betti numbers by current methods requires operating on large exponential-sized matrices (details of TDA and Betti numbers are provided in the next section). The approximation problem solved by our algorithm is believed to be intractable classically (likely belonging to a class of problems called DQC1-hard (Morimae et al., 2014)) under certain settings (Gyurik et al., 2020; Cade & Crichigno, 2021; Crichigno & Kohler, 2022), and in this sense potentially enjoys super-polynomial to exponential speedup over classical algorithms for certain classes of problems (Schmidhuber & Lloyd, 2022). We present a theoretical error analysis for the proposed algorithm, establishing error guarantees for the estimated Betti numbers, and show that the algorithm requires only $\tilde{O}(n/\sqrt{\delta})$-depth circuit complexity. We then present preliminary empirical results from implementations on real hardware and quantum simulations that illustrate the noise resiliency of our algorithm. Our presented theoretical and numerical results demonstrate that NISQ-TDA has the potential to be the *first* generically useful NISQ algorithm.

## 2 PRELIMINARIES

We begin by introducing the key concepts of quantum computing, TDA and quantum TDA (QTDA).

**Quantum computing:** Quantum computing is characterized by operations on the quantum state of $n$ quantum bits or qubits, representing a vector in $2^n$ dimensional complex vector (Hilbert) space. The quantum operations or measurements correspond to multiplying the quantum state vector by certain $2^n \times 2^n$ matrices. Quantum circuits represent these operations in terms of a set of quantum

gates operating on the qubits. The number of these gates and the depth of the circuit define the circuit complexity of a given quantum algorithm. Quantum computers are difficult to build (preparing and maintaining the quantum states is extremely hard) and are very noisy. Therefore, the principles of quantum error correction were proposed to protect the quantum system from information loss and other damages (Gottesman, 2010). A (large-scale) quantum computer with many qubits is said to be fault-tolerant if the device is capable of such quantum error correction. However, realization of such fault-tolerant quantum systems is likely several years away. Currently available quantum computers are termed "Noisy Intermediate-Scale Quantum" (NISQ) (Preskill, 2018), and these devices are prone to considerable error rates and are limited in size by the number of logical qubits available in the system. In order to obtain results with reasonable accuracies on a NISQ device, the quantum circuit implementing a given algorithm needs to be of short depth.

**Topological data analysis:** TDA represents one of the few data analysis methodologies that can process high-dimensional datasets and reduce them to a small set of local and global signature values that are interpretable and laden with predictive and analytical value. Given a set of $n$ data-points $\{x_i\}_{i=0}^{n-1}$ in some space together with a distance metric $\mathcal{D}$, a Vietoris-Rips (Ghrist, 2008) simplicial complex is constructed by selecting a resolution/grouping scale $\varepsilon$ that defines the "closeness" of the points with respect to the distance metric $\mathcal{D}$, and then connecting the points that are a distance of $\varepsilon$ from each other (i.e., connecting points $x_i$ and $x_j$ whenever $\mathcal{D}(x_i, x_j) \leq \varepsilon$, forming a so-called 1-skeleton). A $k$-simplex is then added for every subset of $k+1$ data-points that are pair-wise connected (i.e., for every $k$-clique, the associated $k$-simplex is added).

Let $S_k$ denote the set of $k$-simplices in the Vietoris–Rips complex $\Gamma = \{S_k\}_{k=0}^{n-1}$, with $s_k \in S_k$ written as $\{j_0, \ldots, j_k\}$ where $j_i$ is the $i$th vertex of $s_k$. Let $\mathcal{H}_k$ denote an $\binom{n}{k+1}$-dimensional Hilbert space, with basis vectors corresponding to each of the possible $k$-simplices (all subsets of size $k+1$). Further let $\tilde{\mathcal{H}}_k$ denote the subspace of $\mathcal{H}_k$ spanned by the basis vectors corresponding to the simplices in $S_k$, and let $|s_k\rangle$ denote the basis state corresponding to $s_k \in S_k$. Then, the $n$-qubit Hilbert space $\mathbb{C}^{2^n}$ is given by $\mathbb{C}^{2^n} \cong \bigoplus_{k=0}^{n} \mathcal{H}_k$. The boundary map (operator) on $k$-dimensional simplices $\partial_k : \mathcal{H}_k \to \mathcal{H}_{k-1}$ is a linear operator defined by its action on the basis states as follows:

$$\partial_k |s_k\rangle = \sum_{l=0}^{k-1} (-1)^l |s_{k-1}(l)\rangle, \tag{1}$$

where $|s_{k-1}(l)\rangle$ is the *lower* simplex obtained by leaving out vertex $l$ (i.e., $s_{k-1}$ has the same vertex set as $s_k$ except without $j_l$), and $s_{k-1}$ is $k-1$-dimensional, a dimension less than $s_k$. The factor $(-1)^l$ produces the *oriented* (Ghrist, 2008) sum of boundary simplices, which keeps track of neighbouring simplices so that $\partial_{k-1} \partial_k |s_k\rangle = 0$, given that the boundary of the boundary is empty.

The boundary map $\tilde{\partial}_k : \tilde{\mathcal{H}}_k \to \tilde{\mathcal{H}}_{k-1}$ restricted to a given Vietoris–Rips complex $\Gamma$ is given by $\tilde{\partial}_k = \partial_k \tilde{P}_k$, where $\tilde{P}_k$ is the projector onto the space $S_k$ of $k$ simplices in $\Gamma$. The full boundary operator on the fully connected complex (the set of all subsets of $n$ points) is the direct sum of the $k$-dimensional boundary operators, namely $\partial = \bigoplus_k \partial_k$. The *$k$-homology group* is the quotient space $\mathbb{H}_k := \ker(\tilde{\partial}_k)/\operatorname{img}(\tilde{\partial}_{k+1})$, representing all $k$-holes which are not "filled-in" by $k+1$ simplices and counted once when connected by $k$ simplices (e.g., the two holes at the ends of a tunnel count once). Such global structures moulded by local relationships is what is meant by the "shape" of data. The *$k$th Betti Number* $\beta_k$ is the dimension of this $k$-homology group, namely $\beta_k := \dim \mathbb{H}_k$.

These Betti numbers therefore count the number of holes at scale $\varepsilon$, as described above. By computing the Betti numbers at different scales $\varepsilon$, we can obtain the *persistence barcodes/diagrams* (Ghrist, 2008), i.e., a set of powerful interpretable topological features that account for different scales while being robust to small perturbations and invariant to various data manipulations. These stable persistence diagrams not only provide information at multiple resolutions, but they also help identify, in an unsupervised fashion, the resolutions at which interesting structures exist. The *Combinatorial Laplacian*, or Hodge Laplacian, of a given complex is defined as $\Delta_k := \tilde{\partial}_k^\dagger \tilde{\partial}_k + \tilde{\partial}_{k+1} \tilde{\partial}_{k+1}^\dagger$. From the Hodge theorem (Friedman, 1998; Lim, 2019), we can compute the $k$th Betti number as

$$\beta_k := \dim \ker(\Delta_k). \tag{2}$$

Therefore, computing Betti numbers for TDA can be viewed as a rank estimation problem (i.e., $\beta_k = \dim \tilde{\mathcal{H}}_k - \operatorname{rank}(\Delta_k)$). Additional TDA details can be found in Appendix A.2. The problem

of normalized Betti number estimation (BNE) is defined as (Gyurik et al., 2020): Given a set of $n$ points, its corresponding Vietoris–Rips complex $\Gamma$, an integer $0 \leq k \leq n-1$, and the parameters $(\epsilon, \eta) \in (0, 1)$, find the value $\chi_k \in [0, 1]$ that satisfies with probability $1 - \eta$ the condition

$$\left| \chi_k - \frac{\beta_k}{|S_k|} \right| \leq \epsilon, \tag{3}$$

where $|S_k|$ is the the number of $k$-simplices $S_k \in \Gamma$ or $\dim \tilde{\mathcal{H}}_k$, the dimension of the Hilbert space spanned by the set of $k$-simplices in the complex.

**Quantum TDA:** Lloyd et al. (2016) proposed Quantum TDA (QTDA), an algorithm for solving an approximation of TDA in polynomial time for a class of simplicial complexes. Recent works have shown, e.g., (Gyurik et al., 2020; Schmidhuber & Lloyd, 2022), that the problem QTDA solves approximately is intractable classically for certain classes of complexes. The TDA problem of computing Betti numbers exactly has been shown to be intractable for even quantum computers as decision clique homology has been proven to be QMA1-hard (Crichigno & Kohler, 2022) for clique complexes; and promise weighted clique homology has been shown to be QMA1-hard and contained in QMA (King & Kohler, 2023). The approximative version that QTDA actually solves involves a different computational class: DQC1-hard. This normalized Betti number estimation problem has been shown to be DQC1-hard for general chain complexes (Cade & Crichigno, 2021) and is conjectured to hold for clique complexes (Cade & Crichigno, 2021; King & Kohler, 2023).

QTDA involves two main steps, namely: (a) repeatedly constructing the simplices in the given simplicial complex as a mixed quantum state using Grover's search algorithm (Boyer et al., 1998); and (b) projecting this onto the eigenspace of $\Delta_k$ in order to calculate the *Betti numbers* of the complex, using quantum phase estimation (QPE) (Nielsen & Chuang, 2010) (details are provided in the Appendix A). The computational complexity is $O(n^5/(\delta_k \sqrt{\zeta_k}))$ where $n$ is the number of data points, $\delta_k$ denotes the smallest nonzero eigenvalue of $\Delta_k$, and $\zeta_k$ is the fraction of all simplices of order $k$ in the given complex, resulting in significant speedup over known classical algorithms. However, QTDA requires long-lasting quantum coherence to store the loaded data for the length of the long-depth circuits thus requiring fault-tolerant quantum computing. In particular, Grovers and QPE require precise phase information where any errors would accumulate multiplicatively.

## 3   NISQ-TDA

We now present our proposed quantum algorithm, NISQ-TDA, for estimating the (normalized) Betti numbers of datasets (simplicial complexes) defined through vertices and edges. The algorithm involves three key components, namely: (a) an efficient representation of the full boundary operator as a sum of Pauli operators; (b) a quantum rejection sampling technique to project onto the data-defined simplicial complex; and (c) a stochastic rank estimation method to estimate the output signature *Betti* numbers. In order to calculate the Betti numbers, the first of two major tasks is to construct a quantum circuit that applies the data-defined Laplacian to *any* input set of simplices. In our algorithm, this involves three main sub-components.

The first is a quantum representation of the complete (not data-defined) boundary map operator (say $B$), called the **Fermionic boundary operator** (Cade & Crichigno, 2021; Akhalwaya et al., 2022). It acts on all possible simplices with $n$ points and returns their corresponding boundary simplices. The representation involves only unitary operators written as a sum of Pauli (fermionic) operators. The Hermitian boundary operator $B$ is written as $B = \sum_{i=0}^{n-1} a_i + a_i^\dagger$, where the $a_i$ are the Jordan-Wigner (Jordan & Wigner, 1928) Pauli embeddings corresponding to the $n$-spin fermionic annihilation operators. The implementation of this fermionic boundary operator $B$ on a quantum computer requires only $n$ qubits, $O(n^2)$ gates, and an $O(n)$-depth circuit; see Appendix B for details.

The second sub-component, which we call **Projection onto simplices**, consists of projecting onto the simplicial complex ($\Gamma$) corresponding to the given data by implementing the projector ($P_\Gamma$) using multi-qubit gates and auxiliary flag-registers. A series of multi-control-NOT gates, one for each edge in the complement of the $\varepsilon$-close graph (precomputed classically), checks if the edges of the input simplices (in superposition) are not present in the data. The result is stored in a *flag* register which can either be measured (causing a collapse) or used reversibly allowing for uncomputation. Since there are $\binom{n}{2} \sim O(n^2)$ potential edges, this seems to require $O(n^2)$ depth. However the checks can be run in parallel and in batches using a round-robin procedure for a depth of $O(n)$.

The result of the checks on a maximum of $n/2$ pairs of vertices at a time, needs $n/2$ flag qubits in $n - 1$ rounds, thereby covering all possible $\binom{n}{2}$ pairs of vertices not $\varepsilon$-close. The $n/2$ pairs are chosen such that the C-C-NOT (Toffoli) gates, controlling on pairs of vertex qubits, targeting the flag register, are executed in parallel.

If the flag registers are measured, then just $n/2$ flag qubits suffice in total and can be reused using the power of mid-circuit measure and reset. In each round, we measure the flag register and proceed only if we receive all zeros. This collapses the simplex superposition into those simplices that only have pairs which are not missing from the adjacency graph.

If the flag registers are not measured due to the need for reversibility (which is our case when performing qubitization), then the auxiliary qubits cannot be measured and reused and $O(n^2)$ independent auxiliary qubits are needed.

The 'all-orders' data-defined **Laplacian** can now be expressed as $\Delta = P_\Gamma B P_\Gamma B P_\Gamma$.

Although this simple linear-depth circuit implementation of $P_\Gamma$ suggests a requirement of quantum computers with all-to-all connectivity (as used in our experiments), we can indeed implement it on quantum computers with only linear qubit connectivity using a sorting network approach in $O(n)$ depth (Beals et al., 2013; O'Gorman et al., 2019). The network uses nearest-neighbor SWAP gates and with $n$ layers of such 'qubit swaps', all $\binom{n}{2}$ pair of qubits become nearest-neighbors at some layer; see O'Gorman et al. (2019) for details.

Most importantly, the ability to write the Laplacian in terms of a circuit that does not require accessing stored quantum data is one of the key enabling innovations of NISQ-TDA. The input edge data is not stored on the quantum computer but enters through the presence or absence of the multi-qubit control gates of the projector. Every time the complex projection is called, the data is freshly and accurately injected into the quantum computer. This suggests that NISQ-TDA is partially self-correcting, and under noise presence, the last application of $P_\Gamma$ mitigates the noise. When noise-levels only allow for one coherent application of $P_\Gamma$, this application meaningfully represents the data and can be used for alternate machine learning tasks.

The third sub-component, which we call **Projection to a simplicial order**, is the construction of the projector ($P_k$) onto the $k$-simplex subspace. The circuit is a sequence of control-'add one' sub-circuits that conditions on each vertex qubit of the simplex register and increments a $\log(n)$-sized count register. The operation is equivalent to implementing conditional-permutation, and can be efficiently implemented using diagonalization (Shende et al., 2006) in the Fourier basis. Similarly to $P_\Gamma$, the projection can be executed by measurement collapse (by measuring the count register) or reversibly, enabling uncomputation. The cost in depth is $O(n)$, since each vertex qubit must take a turn to control onto a count register qubit. The data-defined Laplacian corresponding to simplicial order $k$ can thus be written as $\Delta_k = P_k \Delta P_k$.

The second major part of the NISQ-TDA algorithm, which we call the **Stochastic Chebyshev method**, consists of using the above quantum circuit in a larger classically controlled framework, making NISQ-TDA a hybrid quantum-classical algorithm. The classical framework is a stochastic *rank* estimation using the *Chebyshev* polynomials (Ubaru & Saad, 2016; Ubaru et al., 2017). Once we obtain the rank of the Laplacian, we have the Betti numbers $\beta_k = \dim(\ker(\Delta_k)) = |S_k| - \text{rank}(\Delta_k)$, where $S_k \subseteq \Gamma$ is the set of $k$-simplices in the given complex $\Gamma$. Stochastic rank estimation recasts the eigen-decomposition problem into the estimation of the matrix function trace.

Assuming the smallest nonzero eigenvalue of $\tilde{\Delta}_k = \Delta_k/n$ is greater than or equal to $\delta$, we have

$$\text{rank}(\Delta_k) \stackrel{def}{=} \text{trace}(h(\tilde{\Delta}_k)), \text{ where } h(x) = \begin{cases} 1 & \text{if } x > \delta \\ 0 & \text{otherwise} \end{cases}.$$

Supposing $\tilde{\Delta}_k = \sum_i \lambda_i |u_i\rangle\langle u_i|$ is the eigen-decomposition, we have $h(\tilde{\Delta}_k) = \sum_i h(\lambda_i)|u_i\rangle\langle u_i|$, where the step function $h(\cdot)$ takes a value of 1 above the threshold $\delta > 0$ and the eigenvalues of $\tilde{\Delta}_k$ are in the interval $\{0\} \cup [\delta, 1]$. Next, $h(\tilde{\Delta}_k)$ is approximated using a truncated Chebyshev polynomial series (Trefethen, 2019) as $h(\tilde{\Delta}_k) \approx \sum_{j=0}^{m} c_j T_j(\tilde{\Delta}_k)$, where $T_j(\cdot)$ is the $j$th-degree Chebyshev polynomial of the first kind and $c_j$ are the coefficients with closed-form expressions. The trace is approximated using the stochastic trace estimation method (Hutchinson, 1990) given by $\text{trace}(A) \approx \frac{1}{n_v} \sum_{l=1}^{n_v} \langle v_l | A | v_l \rangle$, where $|v_l\rangle, l = 1, \ldots, n_v$, are random vectors with zero mean and uncorrelated coordinates. It can be shown that a set of random columns of the Hadamard matrices works well as

a choice for $|v_l\rangle$, both in theory and practice (see the supplementary material). Sampling a random Hadamard state vector in a quantum computer can be conducted with a short-depth circuit. Given an initial state $|0\rangle$, we randomly flip the $n$ qubits (by applying a NOT gate as determined by a random $n$-bit binary number generated classically). Thereafter, we apply the $n$-qubit Hadamard gate to produce a state corresponding to a random column of the $2^n \times 2^n$ Hadamard matrix. Therefore, the rank of $\Delta_k$ can be approximately estimated as $\text{rank}(\Delta_k) \approx \frac{1}{n_v} \sum_{l=1}^{n_v} \left[ \sum_{j=0}^{m} c_j \langle v_l | T_j(\tilde{\Delta}_k) | v_l \rangle \right]$, where the $c_j$ are Chebyshev coefficients for approximating the step function. Given a circuit that block-encodes $\tilde{\Delta}_k$, we can block-encode a $j$-degree Chebyshev polynomial $T_j(\tilde{\Delta}_k)$ using the idea of qubitization (Low & Chuang, 2019; Gilyén et al., 2019). Details are given in Appendix B.

**NISQ-TDA Algorithm:** We now have all the ingredients to present our NISQ-TDA algorithm:

---

**Algorithm 1** NISQ-TDA Algorithm

---

**Input:** Pairwise distances of $n$ data points and encoding of the $\varepsilon$-close pairs; parameters $\epsilon, \delta$, and $n_v = O(\epsilon^{-2})$; and $n_v$ $n$-bit random binary numbers.

**Output:** Betti number estimates $\chi_k$, $k = 0, \ldots, n-1$.

**for** $l = 1, \ldots, n_v = O(\epsilon^{-2})$ **do**

    **for** $j = 0, \ldots, m = O(\log(1/\epsilon))$ **do**

        **1.** Prepare a random Hadamard state vector $|v_l\rangle$ from $|0\rangle$ using the $l$-th random number.

        **2.** Use the circuits for $P_k$, $P_\Gamma$, and $\tilde{B} = B/\sqrt{n}$ to compute

        $|\phi_l\rangle = |0^q\rangle \tilde{\Delta}_k |v_l\rangle + \left| \tilde{\perp} \right\rangle$, where $q = \#$auxiliary qubits needed for projections.

        **3.** Use qubitization to form: $\left| \psi_l^{(j)} \right\rangle = \left| 0^{q+1} \right\rangle T_j(\tilde{\Delta}_k) |v_l\rangle + |\perp\rangle$ from $|\phi_l\rangle$.

        **4.** Compute the Chebyshev moments $\theta_l^{(j)} = \langle v_l | T_j(\tilde{\Delta}_k) | v_l \rangle$ from $\left| \psi_l^{(j)} \right\rangle$.

    **end for**

    For $j = 0$, estimate $|S_k|$ using the average norm of the $P_\Gamma P_k |v_l\rangle$.

**end for**

Estimate $\chi_k = 1 - \frac{1}{n_v} \sum_{l=1}^{n_v} \left[ \sum_{j=0}^{m} c_j \theta_l^{(j)} \right]$.

*Repeat* for $k = 0, \ldots, n-1$.

---

**Analyses:** Our *NISQ-TDA* algorithm returns the estimates $\chi_k$ for the normalized Betti numbers $\beta_k/|S_k|$, for each order $k = 0, \ldots, n-1$, where $|S_k|$ is the number of $k$-simplices in the given $\Gamma$. We discuss potential scientific machine learning and AI applications of NISQ-TDA in the Appendix. The remainder of this section focuses on theoretical analyses of our *NISQ-TDA* algorithm, with the formal details and proofs provided in Appendix C. We begin with the following main result.

**Theorem 1.** *Assume we are given the pairwise distances of any $n$ data points and the encoding of the corresponding $\varepsilon$-close pairs, together with an integer $0 \le k \le n-1$ and the parameters $(\epsilon, \delta, \eta) \in (0, 1)$. Further assume the eigenvalues of the scaled Laplacian $\tilde{\Delta}_k$ are in the interval $\{0\} \cup [\delta, 1]$, and choose $n_v$ and $m$ such that*

$$n_v = O\left( \frac{\log(1/\eta)}{\epsilon^2} \right) \qquad and \qquad m > \frac{\log(1/\epsilon)}{\sqrt{\delta}}.$$

*Then, the Betti number estimation $\chi_k \in [0, 1]$ by NISQ-TDA, with probability at least $1 - \eta$, satisfies*

$$\left| \chi_k - \frac{\beta_k}{|S_k|} \right| \le \epsilon.$$

Our analysis accounts for errors due to (a) polynomial approximation of the step function; (b) stochastic trace estimator; and (c) also shot noise, i.e., errors in Chebyshev moments estimation and their propagation in classical computation; for details, see Appendix C.

We next discuss the circuit and computational complexities of our proposed algorithm and show that it is NISQ implementable under certain conditions, such as the requirement for simplices-dense complexes, which commonly occur for large resolution scale. The main quantum component of the algorithm comprises the computation of $\theta_l^{(j)} = \langle v_l | T_j(\tilde{\Delta}_k) | v_l \rangle$, for $j = 0, \ldots, m \sim O(\log(1/\epsilon)/\sqrt{\delta})$,

with $n_v \sim O(\epsilon^{-2})$ random Hadamard vectors. The random Hadamard state preparation requires $n$ single-qubit Hadamard gates in parallel and $O(1)$ time. For a given $k$, constructing $\tilde{\Delta}_k$ involves implementing the boundary operator $\tilde{B}$ and the projectors $P_\Gamma$ and $P_k$. The operator $B$, involving the sum of $n$ Pauli operators, can be implemented using a circuit with $O(n)$ gates. $P_k$ requires $O(n \log^2 n)$ gates, while $P_\Gamma$ requires $O(n^2)$ gates, but with both fitting within $O(n)$ depth due to parallelization. Our implementation of $P_\Gamma$ requires $n/2$ auxiliary qubits for measured projection but $O(n^2)$ auxiliary qubits when uncomputation is needed.

The total time complexity of our algorithm is

$$O\left( \frac{1}{\epsilon^2} \max \left\{ \frac{n \log(1/\epsilon)}{\sqrt{\delta}}, \frac{n}{\zeta_k} \right\} \right).$$

Supposing $\delta_k$ is the spectral gap of $\Delta_k$ and $\tilde{\Delta}_k = \frac{\Delta_k}{n}$, then $\delta = \frac{\delta_k}{n}$. The best-known classical algorithm for Betti number estimation of order $k$ has a time complexity of $O(\mathrm{poly}(n^k))$ (Gyurik et al., 2020) or $O(n^{1/\delta \log(1/\epsilon)})$ (Apers et al., 2022). Thus, the QTDA algorithms can achieve superpolynomial to exponential speedups over the best-known classical algorithms whenever we have:

- **Simplices/Clique dense complexes** – the given complex $\Gamma$ is simplices/clique dense, i.e., $\zeta_k$ is large or $|S_k| \sim O(\mathrm{poly}(n))$;

- $O(1/\mathbf{poly}(n))$ **spectral gap** – the spectral gap between zero and nonzero eigenvalues of $\Delta_k$ is not exponentially small, i.e., $\delta$ of $\tilde{\Delta}_k$ is $O(1/\mathrm{poly}(n))$ (Apers et al., 2022); and

- **Large Betti number** – the Betti number $\beta_k$ (and the ratio $\beta_k/|S_k|$) needs to be large so that a large $\epsilon$ suffices to estimate it to a reasonable precision.

A few examples for simplicial complexes that satisfy these conditions are discussed in the Appendix. Further examples and discussions on the potential speedups for quantum TDA algorithms are presented in (Schmidhuber & Lloyd, 2022).

We wish to remark that known examples of simplicial complexes with exponentially many holes (Betti number) are limited. An example family of graphs with exponentially many high-dimensional holes are presented in (Fendley & Schoutens, 2005). More importantly, our algorithm still likely achieves exponential advantage over known classical approaches in efficiently answering the question: *does the given simplicial complex have exponentially many holes or not?* In that regard, our algorithm is indeed applicable to *non-handcrafted high-dimensional classical data*.

From a different point of view, the Chebyshev moments capture the spectral information of $\Delta_k$ and have even more information than the Betti numbers. This therefore opens the door for these ($P_\Gamma$-corrected) noisy moments to be used directly as input features in downstream contexts such as machine learning classification, further relieving the depth and noise requirements of NISQ-TDA.

## 4 EXPERIMENTAL RESULTS

With the theory promising short depths, it remains to demonstrate that NISQ-TDA is sufficiently noise-robust for quantum advantage to be achieved for the actual depths in realizable hardware and under realistic noise levels. Currently optimized classical TDA algorithms cannot compute all Betti numbers for 64 generic vertices (we have empirically verified with a popular public package called *GUDHI* (Maria et al., 2014)). Hence, quantum advantage could possibly be achieved when running NISQ-TDA on 64 vertices. Such large NISQ-TDA circuits are also beyond what is simulatable classically (Pednault et al., 2017; 2019).

We first present the actual depths needed in the form of a depth versus number of vertices plot, which also empirically confirms that circuit depth grows linearly with the number of vertices. Figure 1 shows depths for both actual quantum hardware circuits and generic all-to-all quantum simulator circuits. For the quantum hardware, we employed the public-cloud accessible 'H1' 12-qubit trapped-ion quantum computer from Quantinuum (powered by Honeywell) (Honeywell, 2022). We selected the most conservative number of edges to cover the worst-case depth scenario. The magenta solid points of sub-figure A correspond to Laplacian circuit depths obtained from Quantinuum's own native compiler, and the blue circled points correspond to those obtained from a quantum simulator.

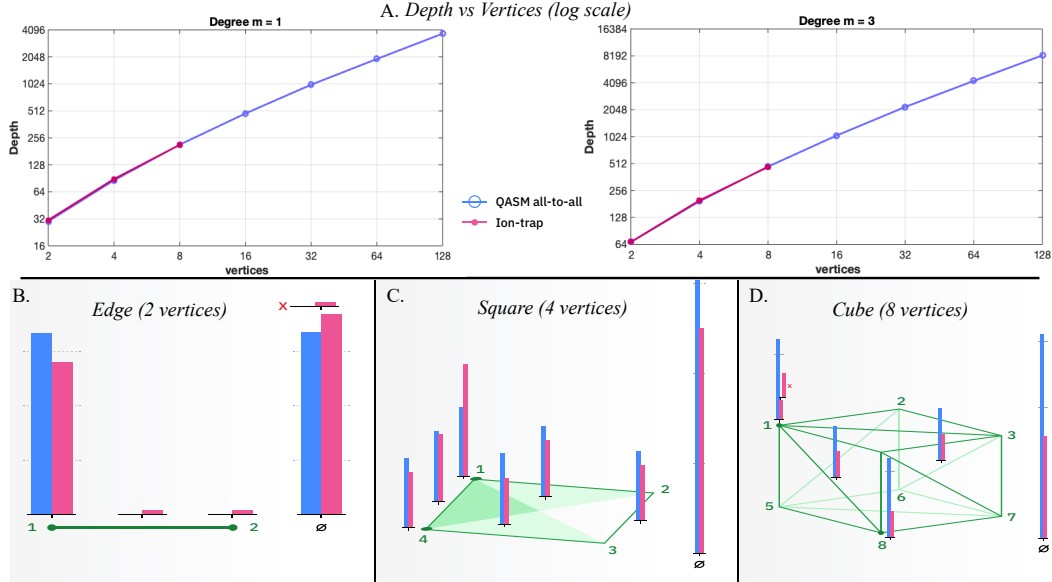

Figure 1: Results from real hardware of Laplacian applications (using measure and reset projections): A. Circuit depth versus the number of vertices for degree $m = 1$ and 3; (B., C. and D.) Histograms of the probability measurements as obtained from the hardware (right, magenta bars) and from a simulator (left, blue bars) for three different datasets namely, an edge (2 vertices), a square (4 vertices), and a cube (8 vertices). $\phi$ defines the null state, and 'X' denotes the probability mass with incorrect flag readings.

We observe that the circuit depth for the Laplacian scales *linearly* with respect to the number of data points. As discussed earlier, our algorithm can be implemented on quantum computers with *linear qubit connectivity* as well, using the sorting network approach (Beals et al., 2013).

The remaining three sub-figures (B, C, D) present the histograms of the top probability measurements for different numbers of vertices (2, 4, 8) for both hardware runs (right, magenta bars) and simulation runs (left, blue bars). These measurements are the raw outputs of the quantum circuit before being converted into expectations (where flag values play a role). The respective complexes chosen correspond to easily understandable shapes (edge, square, cube) represented by the (green) edges. The input simplex set corresponds to a uniform superposition over all simplices (including not shown triangles, tetrahedrons, and all higher-order polytopes). Due to projection onto the specified complexes and interference (correctly eliminating boundaries, sending mass to the null state $\phi$), not all simplices will appear/remain after the application of the Laplacian, demonstrating that the hardware is truly performing a coherent quantum calculation. These sub-figures clearly show that there is agreement between hardware and noise-free simulations on which simplices receive the top probability measurements. Three types of errors are, however, visible by the hardware: reduced probability mass for correct simplices, some small probability mass on incorrect simplices (also not shown are the non-top measurements), and correct simplex mass but incorrect flag readings (marked with a red 'X' in sub-figures B and D). Even with these errors, Figure 1 unequivocally demonstrates that at real-world noise-levels there is sufficient coherence to reproduce the correct interference at the depths of these circuits.

The next task would be to demonstrate that these errors, which inevitably enter, do not dramatically disturb the downstream Betti number calculation. For this we chose complexes with large eigenvalue gaps, and sufficiently many random vectors and shots. The Chebyshev parameters we selected are such that, in the noise-free scenario, the algorithm would calculate the Betti number almost perfectly (i.e., with a mean error of close to zero). Thus, any Betti-number errors involving the noisy simulations are due mainly to the quantum noise and only minimally on downstream classical approximations. In this setup, the error that can naturally be considered tolerable is 0.5, since any error less than 0.5 rounds to exactly the correct Betti number. In Figure 2, we present results from

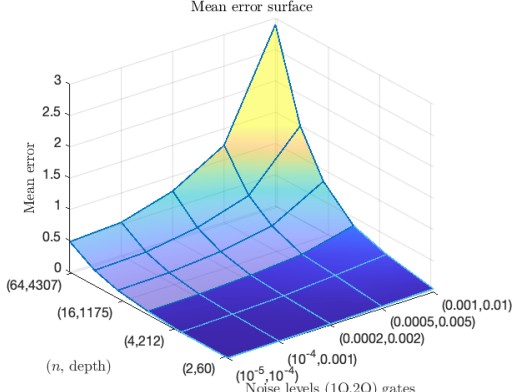
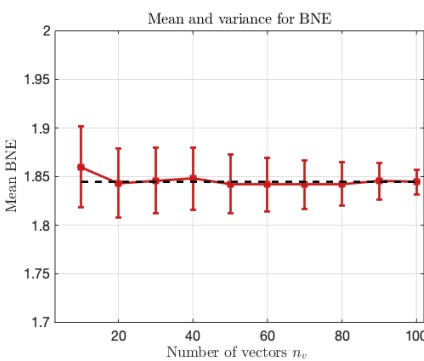

Figure 2: Results from noisy simulations: A. Mean error surface as a function of the noise levels in (1-qubit, 2-qubits) gates and (number of vertices $n$, circuit depth). B. Mean and the variance of the Betti number estimated as a function of the number of random vectors $n_v$ with $n = 8$ vertices, degree $m = 5$ and the noise-level: $(0.001, 0.01)$.

extensive noisy quantum simulations of the non-qubitized version of the algorithm. The right plot shows the mean and the variance (as error bars) of the Betti number estimated as a function of the number of random vectors $n_v$. We note that the mean converges to $\sim 1.84$ (the true Betti number is $\beta_0 = 2$) and, most importantly, the variance reduces as we increase $n_v$. This variance-reduction mitigates errors due to shot noise and randomness in the trace estimation, illustrating the precision-versus-number-of-trials benefit of NISQ-TDA. In the left figure, we present the mean error surface plot for Betti number estimation, as a function of noise levels (chosen triples of measurement, one and two-qubit gate errors) and number of vertices (with concomitant circuit depth). The first number of the listed noise-level pair corresponds to the one-qubit error probability. The measurement and the two-qubit error probabilities are both set to the second value. In the surface plot, the solid region (for $n = 2$ to $n = 8$) corresponds to actual noisy simulations and the translucent region (from 16 to 64 vertices) corresponds to an extrapolation of the surface for larger $n$, which we cannot simulate classically (even $n = 16$ was not simulatable using a large classical machine with 2 GPUs). The surface plot extrapolations provide the minimum noise-level requirements for NISQ-TDA to successfully run on future larger NISQ devices. See Appendix D for additional results, including preliminary results on cosmic microwave background (CMB) data.

## 5 CONCLUSIONS

The true potential of TDA for machine learning has been severely limited because of the computationally prohibitive requirements of classical algorithms. To address this critical issue and revive the potential of TDA as a viable machine learning approach, we presented a new quantum algorithm for Betti number estimation with comprehensive error and complexity analyses. This is one of the *first* quantum machine learning algorithms with short depth and potential significant speedup under certain assumptions. Our algorithm neither suffers from the data-loading problem nor does it likely require fault-tolerant coherence for even mid-size datasets. The algorithm fits the hybrid quantum-classical scheme but within a recently developed randomized-approximation framework. The implementation and successful execution of the entire algorithm on real quantum hardware and noisy simulations was demonstrated, illustrating noise-resiliency at realistic noise-levels. These advantages imply that this algorithm may be one of the few noise-robust quantum algorithms capable of performing an important and useful AI task on near-term (non-fault tolerant) quantum devices, beyond the reach of classical computation. Possible future research directions include: improvements to the algorithm in order to efficiently deploy it on sparsely connected quantum devices; achieving substantial asymptotic speedups under more general settings; and identifying interesting domain problems for which NISQ-TDA can be employed for practical purposes.

ACKNOWLEDGEMENTS

This research was supported in part by the Air Force Research Laboratory (AFRL) grant number FA8750-C-18-0098, and in part by IBM Research, South Africa under the Equity Equivalent Investment Programme (EEIP) of the government of South Africa. VJ is supported in part by the South African Research Chairs Initiative of the National Research Foundation, grant number 78554. Firstly, we wish to thank Brian Neyenhuis, Jennifer Strabley, Chad Edwards, and Tony Uttley from the Quantinuum quantum team for generously providing us credits to access their quantum computers and assisting with the experimentation. Secondly, we would like to thank Adam Connolly and Julien Sorci for pointing out a flaw in the algorithm of an earlier version of the paper. Next, we would like to acknowledge Tal Kachman for the suggestion to use controlled-increment to entangle the simplices with the count register. The authors would also like to thank Scott Aaronson, Paul Alsing, Ryan Babbush, Sergey Bravyi, Chris Cade, Marcos Crichigno, Aram Harrow, Gil Kalai, and Seth Lloyd for valuable discussions.

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

**Topological data analysis on noisy quantum computers
Supplemental material**



Here in the appendix, we present numerous details related to NISQ-TDA, including background information, fleshed out details related to the novel ideas in the proposed algorithm, theoretical analyses and proofs, and additional numerical simulation results.

## A  DETAILS ON QUANTUM ALGORITHMS, TDA, AND QTDA

We begin with a list of desiderata of a quantum algorithm, and technical background details related to TDA and QTDA, including several applications and use cases.

### A.1  DESIDERATA OF A QUANTUM ALGORITHM

An end-to-end algorithm that successfully achieves such quantum advantage on near-term (or even currently available) devices and has practical, real-world and commercial value must arguably satisfy a set of desiderata. Similar to the famous seven DiVincenzo criteria (DiVincenzo, 2000) for *quantum hardware*, we propose a seven-criteria counterpart for *quantum software* that such a quantum algorithm needs to satisfy, as depicted in Figure 3. Although quantum computers can operate on large (exponential) computational spaces, a well-known quantum information bottleneck is the so-called data-loading problem (Aaronson, 2015), in that if the process of loading the data requires exponential time, then the subsequent quantum computational benefits are over-shadowed by this loading time, and do not convey any asymptotic advantage. Indeed, information theory precludes the possibility of lossless exponential compression for generic unstructured data. Therefore, the data size must be small compared to the computational space, and hence, the first criterion is that the algorithm should be effective on *small arbitrary classical input data*. Many quantum machine learning (QML) algorithms (Biamonte et al., 2017; Schuld et al., 2015; Schuld & Killoran, 2019) suffer from this inherent data-loading issue. The term "arbitrary" conveys the desirable property of the general applicability of the algorithm to non-handcrafted classical input data and excludes the data access through an oracle assumptions. A few recent works (Havlíček et al., 2019; Liu et al., 2021; Huang et al., 2022) have demonstrated quantum speedups for different machine learning tasks, but they are restricted to specially tailored classical data, or data from well-defined quantum experiments.

The second requirement of the algorithm is that it should achieve significant asymptotic (ideally exponential) speedup (Knill & Laflamme, 1998; Watrous, 2012) on a provably[1] hard problem. Recently, many hybrid classical-quantum algorithms have been proposed, including variational quantum eigen-solvers (VQE) (Peruzzo et al., 2014) and the quantum approximate optimization algorithm (QAOA) (Zhou et al., 2020), that heuristically exploit the exponential computation space. However, these algorithms do not achieve a provable speedup (Cerezo et al., 2021). Moreover, recent developments of "dequantized" algorithms (Chia et al., 2020; Chepurko et al., 2020; Tang, 2021) have reduced the potential speedups of many of the linear-algebraic QML proposals (Rebentrost et al., 2014; Biamonte et al., 2017; Schuld et al., 2015; Schuld & Killoran, 2019) to be between a nil and at most a modest polynomial.

There are, of course, many quantum algorithms that provably achieve polynomial-to-exponential speedups over the best-known classical methods (Shor, 1994; Grover, 1996; Nielsen & Chuang, 2010; Gilyén et al., 2019). However, the majority of these algorithms require *fault-tolerant* quantum computers (Shor, 1996; Aaronson, 2015; Preskill, 2018), which are error-corrected quantum systems needing a considerably large overhead in resources (number of low-noise qubits and operations) (Arute et al., 2019; Zhao et al., 2020). Fault tolerance has not yet been achieved at scale on currently available quantum devices, and it is likely several years away from full realization. Intriguingly, the qubit numbers, coherence times and noise levels that are currently realized in hardware are not classically simulatable (Pednault et al., 2017; 2019), which raises the question of whether some algorithm could make use of these non-fault-tolerant noisy devices (Noisy Intermediate-Scale Quantum (NISQ) (Preskill, 2018)) for quantum advantage. Hence, the third and fourth requirements of the ideal algorithm are: it should compile to a *short-depth circuit*, allowing execution within

---

[1]By "provably" we mean the problem is *provably* as hard as a computational class that is widely accepted to be classically hard (Watrous, 2012).

**Desiderata of a quantum algorithm**

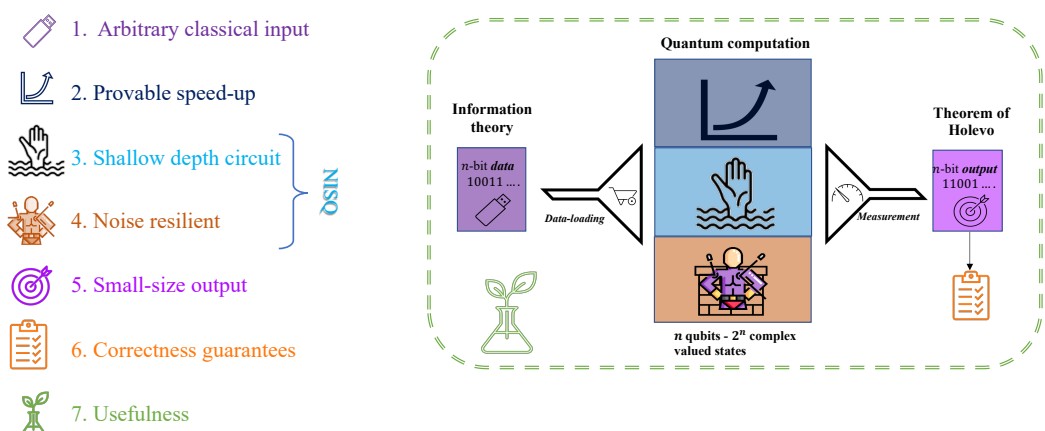

1. Arbitrary classical input
2. Provable speed-up
3. Shallow depth circuit
4. Noise resilient
5. Small-size output
6. Correctness guarantees
7. Usefulness

Figure 3: The armed quantum computing advantage race: a set of seven criteria that an aspired quantum algorithm should satisfy (left), and the end-to-end quantum computational model (right), where the funnels symbolize the narrow input and output bottlenecks and the boxes represent the other desired properties.

the achievable coherence times, and it should be *noise resilient*, thereby tolerating the inevitably introduced noise with each operation.

The fifth requirement demands that the *output should be small in size*. This represents another information theory bottleneck of quantum computing, where Holevo's theorem (Holevo, 1973) suggests we can only read up to $n$ classical bits from an $n$-qubit device, even though the output quantum state is exponential in size. Many QML and other algorithms suffer also from this issue (Aaronson, 2015). Another desirable property of the output forms our sixth desideratum, namely that the output of the algorithm should have *correctness guarantees*. This could be achieved by algorithms with a certain probability of success followed by a procedure to classically verify the correctness of the output, or by algorithms that converge to the correct solution with *statistical error guarantees*.

The final requirement of the algorithm is that it should solve a *useful problem* of real-world applications. The algorithm should be end-to-end and solve a problem with practical use-cases. Recent quantum advantage results (Arute et al., 2019; Zhong et al., 2020; Madsen et al., 2022) fall short with respect to this crucial requirement.

Next, we discuss how our algorithm fares against the set of seven criteria introduced in this paper that should be satisfied by an aspired quantum algorithm.
*Criterion 1* is satisfied with respect to arbitrary small classical input in that the inputs to our algorithm and the quantum computer are just edges between the data-points, with up to $n/2$ edges considered in parallel, and the input can be any dataset or complex defined by its vertices and edges. Moreover, the data is not stored on the quantum computer explicitly, and is input each time the projector $P_\Gamma$ is applied.
*Criterion 2*, that of NISQ-TDA achieving provable asymptotic speedups is likely satisfied for certain instances and classes of complexes (for which the problem is DQC1-hard) as previously discussed; also refer to the discussions in (Crichigno & Kohler, 2022; Schmidhuber & Lloyd, 2022).[2] Since the NISQ-TDA algorithm only requires the complex to be defined by its vertices and edges, one may

---

[2]We note that recent results (Crichigno & Kohler, 2022; Schmidhuber & Lloyd, 2022) have shown the problem of estimating exact Betti numbers of clique complexes to be hard (QMA-1 and NP hard) even for quantum computers, and thus NISQ-TDA is unlikely to achieve quantum advantage for estimation of exact Betti numbers of clique complexes.

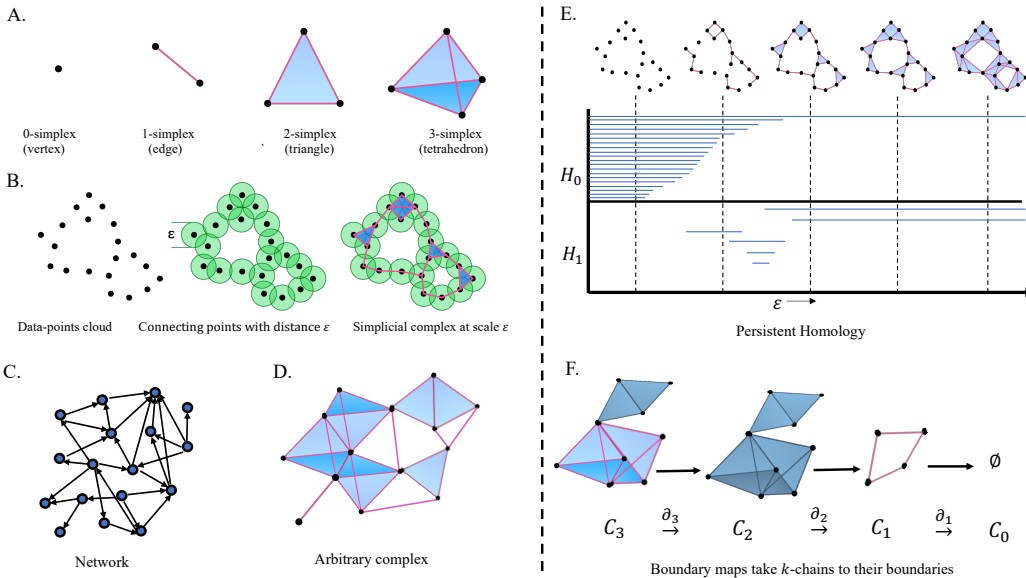

Figure 4: A: $k$-simplices or $k$-chains (fully connected sets of $k+1$ data points) shown for $k = 0, 1, 2, 3$. B: Point cloud of raw data (left); Points can be connected using any arbitrary distance metric $\varepsilon$ (middle), i.e., edges are inserted between points that are within $\varepsilon$ of each other (alternatively, the data could already come with edge information); higher-order $k$-simplices are created for every $k$-clique (right). C. Input data represented as a graph or network, or is given as (D) an arbitrary complex. E: Persistent homology, where the top region shows the edge connections and simplices at different scales, and the bars represent the formation and cessation of connected components ($H_0$) and 2D holes ($H_1$); The number of bars at a given scale $\varepsilon$ equals the Betti numbers $\beta_0$ and $\beta_1$, respectively. F. Chain complex and homology: Sequence of chain groups connected by boundary operators that map $k$-chains to their boundaries.

consider estimation of Betti numbers of other classes of complexes (Schmidhuber & Lloyd, 2022), such as abstract simplicial complexes, Erdos-Renyi complexes and other possible chain complexes. For example, the cube ($n = 8$ instance) considered in our experiments is not a Vietoris-Rips complex (in that case, the cube would have been filled in), but offers one with an example of a complex with a 3D hole with just 8 vertices. Our algorithm should achieve superpolynomial to exponential speedups for complexes with very large number of simplices ($|S_k| \sim O(\text{poly}(n))$), very large Betti number ($\beta_k \approx |S_k|$), and large spectral gap ($1/\sqrt{\delta} \sim O(\sqrt{n})$).

*Criterion 3* is satisfied since the algorithm compiles to a short depth quantum circuit that is implementable on present-day and near-term quantum devices, where the circuit depth will be data dependent (depending on the number of edges, and also on the spectral gap for the polynomial degree). Preliminary experiments show linear scaling of the circuit depths for a fixed polynomial degree.

*Criterion 4* seems to be satisfied as suggested by the preliminary simulation and hardware implementation results, which show that NISQ-TDA is noise-resilient. The noise-resiliency stems from two factors: (a) repeated data loading through the projectors $P_\Gamma$; and (b) the stochastic trace estimator. Each application of $P_\Gamma$ inputs the clean data information afresh, correcting the errors introduced due to noise up to the previous step. The stochastic trace estimator estimates the trace by averaging over many random samples, hence averaging out the noise effects.

*Criterion 5* of small output is naturally satisfied in that the algorithm output is an estimate of the normalized Betti number, and the outputs measured from the quantum computer are the moments.

*Criterion 6* is satisfied, as we presented error guarantees for NISQ-TDA and discussed the conditions under which the algorithm will likely achieve speed up over classical algorithms.

*Criterion 7* is likely satisfied in that it can certainly solve interesting useful problems, such as estimating whether or not a given complex has exponentially many holes and extracting spectral information of high-order Laplacians, with substantial speedups over known classical approaches, even if

it is unclear whether NISQ-TDA will achieve substantial speedups on arbitrary data for Betti number estimation; some specific useful applications are discussed below.

## A.2 Topological Data Analysis

Figure 4 illustrates the key concepts of TDA. A $k$-simplex or a $k$-clique or $k$-chain is a collection of $k + 1$ fully connected points (see Figure 4(A)), and a simplicial complex (clique or chain complex) is a collection of such (nested) simplices (Figure 4(B)). Formally, an abstract simplicial complex $\Gamma$ (Figure 4(D)) is a set of simplices satisfying: (a) if a simplex is in $\Gamma$, then all its faces are in $\Gamma$, and (b) intersection of two simplices in $\Gamma$ is through a face of each of them. A clique complex is typically defined for a graph, and is a simplicial complex with the $k$-simplices given by the $k$-cliques of the graph. A simplicial complex also defines a collection of abelian groups. A chain complex is a sequence of modules (groups) connected by boundary operators (defined later); see Figure 4(F).

Homology provides us with a linear-algebraic tool to extract, from simplicial complexes derived from data, values that describe the shape of the data, such as the number of connected components (clusters), tunnels (e.g., a doughnut shape), holes (as in Swiss cheese cavities), or higher-dimensional voids, together known as the Betti numbers (Ghrist, 2008) (see Figure 4(C)). As the $\epsilon$-distance (Figure 4(B)) is varied the simplicial complex changes, altering the Betti numbers (Figure 4(C)). The pattern of changing Betti numbers (*persistent homology* (Ghrist, 2008)) provides a topological characterization of the data distribution that is scale-independent, invariant under rotation and translation, and robust under variations due to data representation, data sampling, and data noise. These Betti numbers are defined in terms of a Laplacian matrix ($\Delta_k$), which takes a set of simplices and returns an *oriented* sum of all simplices connected via the *boundary* simplices of the input. The signed orientations allow simplices to cancel out if they encompass a hole, where the hole-surrounding boundaries form the *kernel* set of the Laplacian. The Betti numbers are precisely the size of these kernels of various simplicial orders. The Laplacian can be very large for high-order $k$, and classical algorithms for Betti number calculation become intractable even for $k \geq 3$. In this paper, we present a new quantum representation of the Laplacian separating the boundary action from simplicial complex construction enabling implementation on near-term devices.

## A.3 QTDA Algorithm

The seminal approach of Lloyd et al. (2016) to estimate the Betti numbers using quantum computers, which was further analyzed by (Gunn & Kornerup, 2019) and (Gyurik et al., 2020), comprises two main steps. The first step of the algorithm is to create a mixed state $\rho_k$ over the states $|s_k\rangle$ of $k$-simplices (over $\tilde{\mathcal{H}}_k$) that are in the complex $\Gamma$. The second step is to use Hamiltonian simulation (of the boundary operator or the Laplacian $\Delta_k$) and quantum phase estimation (QPE) with $\rho_k$ in the input register (repeatedly projecting simplices from the complex onto the kernel) to estimate the kernel dimension of the Laplacian.

In order to prepare the maximally mixed state $\rho_k$ as part of the first step, the QTDA algorithm first uses Grover's search algorithm (Boyer et al., 1998) to construct the $k$-simplex state

$$|\psi_k\rangle = \frac{1}{\sqrt{|S_k|}} \sum_{s_k \in S_k} |s_k\rangle,$$

for the set $S_k$ with $|S_k| = \dim \tilde{\mathcal{H}}_k$. Then, the mixed state

$$\rho_k = \frac{1}{|S_k|} \sum_{s_k \in S_k} |s_k\rangle \langle s_k|$$

can be prepared from $|\psi_k\rangle$ by applying the CNOT gate to each qubit and tracing out into the auxiliary zero qubits. The time complexity of this step is $O\left(\frac{k^2}{\sqrt{\zeta}}\right)$, where $\zeta := \frac{|S_k|}{\binom{n}{k+1}}$ is the fraction of $k$-simplices that are in the complex $\Gamma$. The number of gates required for this step is $O\left(kn^2 + \frac{nk}{\sqrt{\zeta}}\right)$ (Gunn & Kornerup, 2019). We believe this step is unnecessary, because a random simplex of order $k$ can be drawn from the complex efficiently. Nevertheless, the same Grover's search is needed to restrict the boundary operator to the complex in the next step.

The second step uses QPE to estimate the kernel dimension of the Laplacian $\Delta_k$. For this, the following *Dirac operator* (the square root of the generalized Laplacian)

$$
\tilde{B} = \begin{pmatrix}
0 & \tilde{\partial}_1 & 0 & \cdots & \cdots & 0 \\
\tilde{\partial}_1^\dagger & 0 & \tilde{\partial}_2 & 0 & \cdots & 0 \\
0 & \tilde{\partial}_2^\dagger & 0 & \ddots & \cdots & 0 \\
\vdots & \vdots & \ddots & \ddots & \ddots & \vdots \\
\vdots & \vdots & \vdots & \ddots & 0 & \tilde{\partial}_{n-1} \\
0 & 0 & 0 & \cdots & \tilde{\partial}_{n-1}^\dagger & 0
\end{pmatrix}
\tag{4}
$$

is first simulated such that $\tilde{B}^2 = \text{Blockdiag}[\Delta_1, \ldots, \Delta_n]$ is a block diagonal matrix[3], since $\tilde{\partial}_k \tilde{\partial}_{k+1} = 0$. Given that $\tilde{B}$ has the same nullity (kernel) as $\tilde{B}^2$, the idea is to use Hamiltonian simulation of $\tilde{B}$ (i.e., implement $U = e^{i\tilde{B}}$), and use QPE with $\rho_k$ (computed in the first step) as the input state to estimate its eigenvalues. Since $\tilde{B}$ is an $n$-sparse Hermitian with entries $\{0, \pm 1\}$, it is claimed that this can be simulated using $O(n)$ qubits and $O(n^2)$ gates (Low & Chuang, 2017).[4]

QPE yields an approximate estimate of the eigenvalues of $\Delta_k$. We need to scale $\Delta_k$ such that its spectrum is in the interval $[0, 1]$, in order to avoid multiples of $2\pi$; see Section C for details on scaling. Supposing the smallest nonzero eigenvalue of (the scaled) $\Delta_k$ is greater than $\delta > 0$, we then need to estimate the eigenvalues with a precision of at least $\frac{1}{\delta}$ in order to distinguish an estimated zero eigenvalue from others. Therefore, the time complexity of this step is $O(\frac{n^2}{\delta})$ and requires as many gates for its implementation.

This use of QPE provides us with an approximate estimate of some random eigenvalue of $\Delta_k$. For BNE with additive error $\epsilon$, we need to repeat the two steps $O(\epsilon^{-2})$ times. Hence, the total time complexity of QTDA (original Lloyd et al. (2016) version[5]) for BNE with $\left| \chi_k - \frac{\beta_k}{\dim \tilde{\mathcal{H}}_k} \right| \leq \epsilon$ is given by

$$
O\left( \frac{n^4}{\epsilon^2 \delta \sqrt{\zeta}} \right).
$$

**Remark 1** (Time Complexity Discrepancy). *We note that there is a discrepancy in the total time complexity of the QTDA algorithm reported in (Lloyd et al., 2016) and in the subsequent articles by (Gunn & Kornerup, 2019) and (Gyurik et al., 2020), primarily due to differences in the underlying assumptions. This relates to simulation of the matrix $\tilde{B}$ or $\Delta_k$, where Lloyd et al. (2016) suggest the requirement of constructing and applying the projector $\tilde{P}_k$ at each round (possibly using Grover's search algorithm, although some implementation details are missing). Hence, the total time complexity in (Lloyd et al., 2016) is a product of the time complexity of the two steps. In contrast, the follow-up studies by (Gunn & Kornerup, 2019) and (Gyurik et al., 2020) assume that we have access to $\tilde{B}$ or $\Delta_k$ as an $n$-sparse matrix, in order to simulate it in the second step, and therefore the time complexities of the two main steps are added in (Gunn & Kornerup, 2019; Gyurik et al., 2020) to obtain the total computational time complexity.*

The subsequent articles by (Gunn & Kornerup, 2019) and (Gyurik et al., 2020) do not address the issue of efficient quantum construction of $\tilde{B}$ or $\Delta_k$ from the pairwise distances of the $n$ points, and assume that oracle access is given to the nonzero entries of $\tilde{B}$ and their locations.

### A.4 POTENTIAL USE-CASES

The proposed *NISQ-TDA* method is advantageous for computing the Betti numbers of simplices-dense complexes, especially higher order Betti numbers (the regime where classical methods fail). Here, we briefly discuss some of the potential applications where *NISQ-TDA* can be extremely useful and even revolutionary.

---

[3] The block diagonal form is obtained in the Hamming weight sorted representation of the simplices.

[4] The serious issue is that the restricted Dirac operator is not on hand, and requires $\tilde{P}_k$ to be known; see Remark 1.

[5] Except that we adjust the cost of QPE under our spectral interval assumption.

**Neural networks.** The training and application of artificial Neural Networks (NN) is a key methodology of artificial intelligence, and understanding the processing and capabilities of neural networks is critical to that methodology. TDA has been proposed for the analysis of neural networks; in one such proposal (Naitzat et al., 2020), the dataset points are considered to comprise a sample of a surface and the *topological complexity* $T_0$, considered here to be the sum the Betti numbers, is estimated for the dataset surface. Each layer $\ell_i$ computes a transformation of its input, and so yields a sample of a surface, whose topological complexity $T_i$ can also be estimated. In the analysis of (Naitzat et al., 2020), the resulting sequence $T_0, T_1, \ldots, T_h$, for a trained network with $h$ layers, is found to decrease rapidly, so that network processing can be understood as a sequence of topological simplifications. By considering the nature and rapidity of this simplification, insights into network architecture and processing are obtained. Another application of TDA (Guss & Salakhutdinov, 2018) considers the question of the appropriate width and height of networks, via the estimation of the capacity of a network as a function of the network height and maximum width. Here the network is used for classification, dividing the input space into positive and negative instances, and with the *decision boundary* between these two sets. The capacity of the network is considered to be the maximum topological complexity of the decision boundary determined by the network. TDA can be applied to the analysis of decision boundaries. Most importantly, we note that where both (Naitzat et al., 2020) and (Guss & Salakhutdinov, 2018) consider only small Betti numbers, and mainly low-dimensional datasets, further insights could be obtained via estimation of higher Betti numbers of higher-dimensional data, reachable via NISQ-TDA.

**Cosmic microwave background.** Next, we consider the CMB application, a potential use-case of NISQ-TDA, for further discussion due to three favourable reasons. The first is due to the immense scientific and cosmic value that is to be gained from a meticulous study of the available and prospective high-quality CMB data, including the testing of theories of fundamental physics, the constraining of the fundamental constants of the universe, and even the probing of the existence of parallel and past universes. Secondly, the CMB use-case has been extensively studied from a classical TDA perspective. Indeed, promising results have already been empirically demonstrated for low-order Betti numbers (Cole & Shiu, 2018a; Biagetti et al., 2021). A few papers have even produced convincing theoretical results demonstrating the power of Betti numbers to shed light on the CMB use-case, in particular (Feldbrugge et al., 2019b) have derived analytic expressions for low-order Betti numbers proving their usefulness. Thirdly, (Adler et al., 2014) have proven that low and, more importantly, high-order Betti numbers associated with a random sample of points generated from different probability distributions contain valuable characterizing properties.

We make the novel connection between the results in (Adler et al., 2014) and its application to the study of the CMB. As a proof of principle, we have implemented the insights from their paper and empirically demonstrated distinguishability of the studied distributions, experimenting with a small number of points to match the regime of interest for NISQ-TDA; see section D.2 for these results. The preliminary results suggest the Betti numbers of a small sample set can be used for the detection of non-Gaussianity in the CMB data, and NISQ-TDA can be used for the estimation of these Betti numbers of all order.

**Neuroscience.** Topology-based methods have been used to detect interpretable structures from neural activity and connection data in Neuroscience (Giusti et al., 2015). Such structural features can yield key insights into neurological processes. In (Giusti et al., 2015), clique topology was used to extract invariant features from neural data that reveal geometric structures of neural correlations in the rat hippocampus. In particular, Betti curves, the distribution of the Betti numbers $\beta_k$ as a function of the edge density $\rho$, and integrated Betti values $\bar{\beta}_k$, defined as $\bar{\beta}_k = \int_0^1 \beta_k(\rho)d\rho$, were considered as clique topology features and were computed from neural connectivity data. The article showed that the geometric signatures observed using these topological features in the neural correlations revealed many interesting insights. Neural connections have many unknown non-linearities, and clique topology can extract such nonlinear features present in the connectivity data, presenting novel insights in brain activity and connectivity. Indeed, the clique topology features such as Betti curves and integrated Betti values considered in this application require us to compute the Betti numbers of many clique-dense complexes. Thus far, only small order Betti numbers have been considered due to computational impediments. Higher order clique topology features might reveal many new insights in neural connectivity.

**Genetics.** TDA is also playing an important role in various aspects of genetics. As one representative example, TDA has been exploited to address an important problem in the sequencing of a sample of a collection of genomes en masse when distinct organisms with very similar genomes are present (Guzman-Saenz et al., 2019). This problem arises in the investigation of the community of constituent microorganisms in a micro-environment where differentiating between similar organisms leads to many false positive identifications because of the numerous possible assignments of short sequencing reads to reference genomes. TDA is used to address this problem by extracting information from the geometric structure of data, where said structure is defined by relationships between sequencing reads and organisms in a reference database, resulting in separation of true positives from false positives and capturing the non-obvious structure defined by the reads indiscriminately mapping to multiple organisms (Guzman-Saenz et al., 2019). As another representative example, TDA has been exploited to address an important problem in phenotype prediction as to whether RNA sequencing-based gene expression contains enough information to separate healthy and afflicted individuals (Mandal et al., 2020a). This problem is particularly difficult given the poor phenotype predictions from standard machine learning methods. By taking into account the topological information and features relevant to classification from gene expression data, TDA is used to understand the shape of the very high-dimensional gene expression data and to obtain topological summaries of the gene expressions of subjects contained within this data, rendering a significant improvement in phenotype prediction of disease and confirming that gene expression can be a useful indicator of the presence or absence of a health condition (Mandal et al., 2020a). In both of these examples, only small Betti numbers are considered, whereas significant insights should be obtainable using the higher Betti numbers within reach via NISQ-TDA.

## B NISQ-TDA DETAILS

Here, we present additional details related to different aspects of the proposed algorithm. The first key innovation of NISQ-TDA is the fermionic representation of the full boundary operator $B$.

### B.1 FERMIONIC BOUNDARY OPERATOR

Fermionic fields obey Fermi–Dirac statistics, which means that they admit a mode expansion in terms of creation and annihilation oscillators that anticommute. Exploiting this fact, it is convenient to map Pauli spin operators to fermionic creation and annihilation operators. The Jordan–Wigner transformation (Jordan & Wigner, 1928) is one such mapping. In this section, we will make use of it to express the boundary matrix.

The Lloyd et al. (2016) restricted boundary operator given in (1) is not in a form that can be easily executed on a quantum computer, nor does it act on all orders, $k$, at the same time. In particular, it is a high-level description of the action of the boundary operator on a single generic $k$-dimensional simplex with the location of the ones assumed to be known. Furthermore, this representation is in tensor product form composed of quantum computing primitives that directly map to quantum gates in the quantum circuit model. To begin, define the operator:

$$Q^+ := \frac{1}{2}(\sigma_x + \mathtt{i}\sigma_y) = \begin{pmatrix} 0 & 1 \\ 0 & 0 \end{pmatrix}. \tag{5}$$

This allows writing the *full* boundary operator in terms of the above operator:

$$
\begin{aligned}
\partial^{(n)} &:= \sigma_z \otimes \ldots \otimes \sigma_z \otimes Q^+ \\
&+ \sigma_z \otimes \ldots \otimes \sigma_z \otimes Q^+ \otimes I \\
&\vdots \\
&+ \sigma_z \otimes Q^+ \otimes I \otimes \ldots \\
&+ Q^+ \otimes I \otimes I \otimes \ldots \\
&= \sum_{i=0}^{n-1} a_i,
\end{aligned} \tag{6}
$$

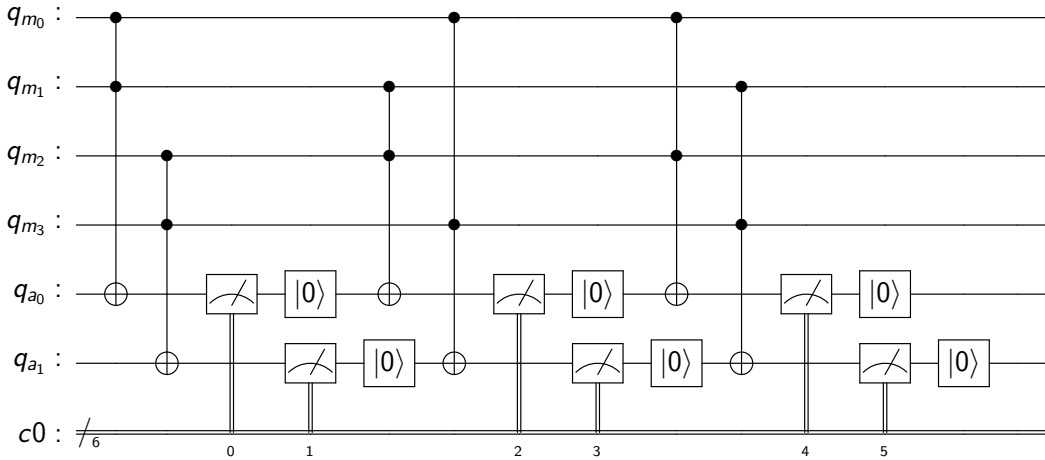

Figure 5: Projection onto the simplicial complex $P_\Gamma$: Example circuit diagram with $n = 4$ vertices (and six edges)

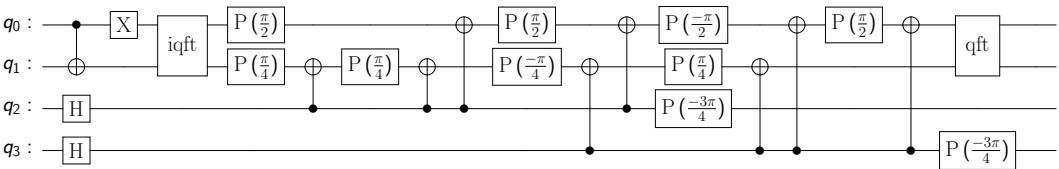

Figure 6: Projection onto the simplicial order $P_k$: Example circuit diagram with $n = 4$ vertices

where the $a_i$ are the Jordan–Wigner (Jordan & Wigner, 1928) Pauli embeddings corresponding to the $n$-spin fermionic annihilation operators. This fermionic boundary map representation was presented in (Cade & Crichigno, 2021) and (Akhalwaya et al., 2022). For details on its correctness and $O(n)$-depth unitary circuit to construct it on a quantum computer, see Akhalwaya et al. (2022).

## B.2 PROJECTION OPERATORS

In the main paper, we discussed the two projection operators, namely, projection onto the simplicial complex $P_\Gamma$ and projection onto the order $P_k$, which are key components of the proposed NISQ-TDA algorithm. Here, we present example circuit diagrams for these two operations. Figure 5 presents an example circuit for $P_\Gamma$ with $n = 4$ vertices, assuming all possible six edges are present. We have $n = 4$ main qubits, and $n/2$ auxiliary qubits needed, along with $n - 1$ measure and reset operations.

As discussed in the main text, we can alternatively use $\binom{n}{2}$ auxiliary qubits, one for each possible edge. Then, we can implement the projector $P_\Gamma$ reversibly. With this approach, we can obtain a block-encoding of the Laplacian $\tilde{\Delta}_k$ using $O(n^2)$ auxiliary qubits. The key advantage of this approach is that block-encoding the Chebyshev polynomial $T_j(\tilde{\Delta}_k)$ is now possible. See details in the next section.

Figure 6 presents a sample circuit for $P_k$, again with $n = 4$ vertices. We use Fourier transform (QFT/iQFT circuits) as a change of basis, followed by the permutation circuit (phase rotations) to implement the count increment.

### B.3 Stochastic Rank Estimation

Another key ingredient of our proposed NISQ-QTDA algorithm is a stochastic rank estimation procedure that estimates the Betti number $\beta_k$ by estimating the rank of $\Delta_k$, which replaces the QPE component of the Lloyd et al. (2016) algorithm. The standard approach to estimate the rank of a square matrix is to compute all of its eigenvalues and count the number of nonzero eigenvalues, for which prior work (Lloyd et al., 2016) has employed QPE. In this paper, we propose a rank estimation procedure that does not require any decomposition of the corresponding matrix. In particular, our rank estimation approach is based on the classical *stochastic Chebyshev* method (Ubaru & Saad, 2016; Ubaru et al., 2017). Namely, the proposed approach recasts the rank estimation problem to one of estimating the trace of a certain (step) function of the matrix. The trace is then approximated by a stochastic trace estimator, where the step function is approximated by a Chebyshev polynomial approximation.

**Stochastic trace estimator:**  Given a Hermitian matrix $A \in \mathbb{R}^{N \times N}$, the stochastic trace estimation method (Hutchinson, 1990; Avron & Toledo, 2011) uses only the moments of the matrix to approximate the trace. In the classical setting, $\mathrm{trace}(A)$ is estimated by first generating random vector states $|v_l\rangle$ with random independent and identically distributed (i.i.d.) entries, $l = 1, .., n_v$, and then computing the average over the moments $\langle v_l | A | v_l \rangle$; namely,

$$\mathrm{trace}(A) \approx \frac{1}{n_v} \sum_{l=1}^{n_v} \langle v_l | A | v_l \rangle. \tag{7}$$

Any random vectors $|v_l\rangle$ with zero mean and uncorrelated coordinates can be used (Avron & Toledo, 2011).

In the quantum setting, however, particularly with NISQ computations, generating random states $|v_l\rangle$ of exponential size with i.i.d. entries is not viable. Alternatively, it has been shown that random columns drawn from the Hadamard matrix work very well in practice for stochastic trace estimation (Fika & Koukouvinos, 2017). Sampling a random Hadamard state vector in a quantum computer is extremely simple and can be conducted with a short-depth circuit. Given an initial state $|0\rangle$, we randomly flip the $n$ qubits (possibly by applying a NOT gate as determined by a random $n$-bit binary number $\in [0, 2^n - 1]$ generated classically). Thereafter, we simply apply Hadamard gates to all qubits. This produces a state corresponding to a random column of the $2^n \times 2^n$ Hadamard matrix. The columns of a Hadamard matrix have pairwise independent entries. Hence, we consider the random state vector $|v_l\rangle = |h_{c(l)}\rangle$, i.e., some random Hadamard column with $c(l)$ defining the random index, and then we estimate the moments $\langle h_{c(l)} | A | h_{c(l)} \rangle$ and average over the $n_v$ samples to approximate the trace. The error analysis for this approach is presented in the next section.

Alternatively, quantum t-design circuits are a popular way to generate pseudo-random states (Ambainis & Emerson, 2007; Brakerski & Shmueli, 2019). A t-design circuit outputs a state that is indistinguishable from states drawn from a random Haar measure. These t-designs in a quantum computer are equivalent to $t$-wise independent vectors in the classical world (Ambainis & Emerson, 2007). Short-depth circuits exist (though not as short as above) that are approximate $t$-designs (Brakerski & Shmueli, 2019). Such $t$-design circuits can be used to generate the random states $|v_l\rangle$ for trace estimation. Indeed, random vectors with just 4-wise independent entries suffice for trace estimation (we omit the details here because this approach is less competitive than the above super-short-depth Hadamard construction).

**Chebyshev approximation:**  Assuming the smallest nonzero eigenvalue of $A$ is greater than or equal to $\delta$, then the rank of $A$ can be written as

$$\mathrm{rank}(A) \stackrel{def}{=} \mathrm{trace}(h(A)), \text{ where } h(x) = \left\{ \begin{array}{ll} 1 & \text{if } x > \delta \\ 0 & \text{otherwise} \end{array} \right. . \tag{8}$$

Given the eigen-decomposition $A = \sum_i \lambda_i |u_i\rangle\langle u_i|$, we have the matrix function $h(A) = \sum_i h(\lambda_i)|u_i\rangle\langle u_i|$ where the step function $h(\cdot)$ takes a value of 1 above the threshold $\delta > 0$. The parameter $\delta$ is assumed to be known (or, in the classical setting, can be estimated using the spectral density method (Ubaru & Saad, 2016)). In the case of TDA, for many simplicial-complex types, a lower bound for the smallest nonzero eigenvalue of $\Delta_k$ can be estimated; refer to Section C.3 for a few examples.

Next, the approach of Ubaru et al. (Ubaru & Saad, 2016; Ubaru et al., 2017) consists of approximating the matrix function $h(A)$ by employing Chebyshev polynomials (Trefethen, 2019), and estimating the trace using the stochastic estimator (7). More specifically, $h(A)$ is approximately expanded in the following manner

$$h(A) \approx \sum_{j=0}^{m} c_j T_j(A),$$

where $T_j(x)$ is the $j$th-degree Chebyshev polynomial of the first kind, formally defined as $T_j(x) = \cos(j \cos^{-1}(x))$. We therefore have $T_0(x) = 1$, $T_1(x) = x$ and $T_{j+1}(x) = 2x T_j(x) - T_{j-1}(x)$. The expansion coefficients $c_j$ for the polynomial to approximate a step function $h(t)$, taking value 1 in $[a, b]$ and 0 elsewhere, are known to be given by

$$c_j = \begin{cases} \frac{1}{\pi}(\cos^{-1}(a) - \cos^{-1}(b)) & : j = 0 \\ \frac{2}{\pi}\left(\frac{\sin(j\cos^{-1}(a)) - \sin(j\cos^{-1}(b))}{j}\right) & : j > 0 \end{cases}.$$

Therefore, the rank of a given matrix $A$, with the smallest nonzero eigenvalue greater than or equal to $\delta$, can be approximately estimated using the stochastic Chebyshev method as:

$$\text{rank}(A) \approx \frac{1}{n_v} \sum_{l=1}^{n_v} \left[ \sum_{j=0}^{m} c_j \langle v_l | T_j(A) | v_l \rangle \right]. \tag{9}$$

The method estimates the rank using only the Chebyshev moments of the matrix $\langle v_l | T_j(A) | v_l \rangle$. Classically, these moments are typically built using the three-term recurrence (Ubaru & Saad, 2016). Therefore, we need to compute the Chebyshev moments $\langle v_l | T_j(\tilde{\Delta}_k) | v_l \rangle$ for $j = 0, \dots, m$ on the quantum computer using qubitization[6].

**Qubitization - Block encoding Chebyshev polynomial of a Hermitian matrix:** Suppose we are given a $(1, q)$-block encoding $U_A$ of a Hermitian matrix $A$ (Lin, 2022). Then we can use the qubitization idea (Low & Chuang, 2019; Gilyén et al., 2019) to obtain a block encoding of $T_j(A)$.

Given the eigen-decomposition $A = \sum_i \lambda_i |u_i\rangle\langle u_i|$, we have for any eigenstate $|u_i\rangle$ that

$$U_A |0^q\rangle |u_i\rangle = |0^q\rangle A |u_i\rangle + \left|\tilde{\perp}_i\right\rangle = \lambda_i |0^q\rangle |u_i\rangle + \left|\tilde{\perp}_i\right\rangle,$$

where $\left|\tilde{\perp}_i\right\rangle$ is an unnormalized state that is orthogonal to all states of the form $|0^m\rangle |\psi\rangle$. From above, we have $\left|\tilde{\perp}_i\right\rangle = \sqrt{1 - \lambda_i^2} |\perp_i\rangle$ for a normalized state $|\perp_i\rangle$. If $U_A$ is Hermitian, we can show that $\mathcal{H}_i = span(|0^q\rangle |u_i\rangle, |\perp_i\rangle)$ is an invariant subspace of $U_A$.

Next, we consider a projection operator to the basis $\mathcal{B}_i = span(|0^q\rangle |u_i\rangle, |\perp_i\rangle)$ such that:

$$[Z_\Pi]_{\mathcal{B}_i} = \begin{bmatrix} 1 & 0 \\ 0 & -1 \end{bmatrix}.$$

That is, $Z_\Pi$ is as a reflection operator restricted to each subspace $\mathcal{H}_i$. Next, if we define a rotation matrix

$$O = U_A Z_\Pi,$$

then $\mathcal{H}_i$ is invariant to this matrix and its powers. We can show that

$$O^j = \begin{bmatrix} T_j(A) & * \\ * & * \end{bmatrix}.$$

Namely, $O^j = (U_A Z_\Pi)^j$ is a $(1, q)$-block encoding of $T_j(A)$.

---

[6]A previous version of this paper proposed an algorithm that did not make use of qubitization but instead used the relationship between the Chebyshev polynomials and the moments of the Laplacian. However this approach incurs an additional complexity factor (see Table C.3) as pointed out to us by Adam Connolly and Julien Sorci, necessitating the use of qubitization to remove this overhead.

For $q = 1$, $Z_\Pi$ is just a Pauli $Z$ gate (defined on the subspace of the projection). For $q > 1$, $Z_\Pi$ can be implemented using one additional qubit, two control gates, and a Pauli $Z$ gate. For details, see Chapter 7 in (Lin, 2022). Therefore, given a Hermitian block encoding of $A$, we can compute a $(1, q + 1)$-block encoding of $T_j(A)$ using qubitization.

In our case, if we use an $\binom{n}{2}$ auxiliary approach for $P_\Gamma$, then using the circuits for $P_k$, $P_\Gamma$ and $\tilde{B} = B/\sqrt{n}$, we can compute a $(1, q)$-Hermitian block encoding of $\tilde{\Delta}_k$, with $q = O(n^2)$, and form

$$|\phi_l\rangle = |0^q\rangle \, \tilde{\Delta}_k \, |v_l\rangle + \left| \tilde{\perp} \right\rangle .$$

Then, using the above approach, we can obtain a $(1, q + 1)$-block encoding of $T_j(\tilde{\Delta}_k)$, and form

$$\left| \psi_l^{(j)} \right\rangle = |0^{q+1}\rangle \, T_j(\tilde{\Delta}_k) \, |v_l\rangle + |\perp\rangle$$

from $|\phi_l\rangle$. We can next compute the Chebyshev moments $\theta_l^{(j)} = \langle v_l | \, T_j(\tilde{\Delta}_k) \, |v_l\rangle$ from $\left| \psi_l^{(j)} \right\rangle$ for $j = 0, \ldots, m$ and $l = 1, \ldots, \mathrm{n_v}$. If we use the measure-and-reset approach for $P_\Gamma$ ($n/2$ auxiliary and $n - 1$ measure and resets), then implementing $Z_\Pi$ will be more involved.

The next section presents our analysis of the error and the computational complexities of the above NISQ-TDA algorithm.

## C  THEORETICAL ANALYSES

We turn to the theoretical analysis of our proposed NISQ-QTDA algorithm, first discussing the error analysis that provides bounds on the number of random vectors $\mathrm{n_v}$ and the polynomial degree $m$ needed to achieve BNE with $\left| \chi_k - \frac{\beta_k}{|S_k|} \right| \leq \epsilon$, and subsequently presenting the gate and time complexities of the algorithm. We then discuss different scenarios under which the QTDA algorithms can achieve significant speedups over classical algorithms, including when our proposed algorithm can be NISQ implementable.

### C.1  ERROR ANALYSIS

Algorithm 1 returns a Betti number estimate $\chi_k$ for each order $k = 0, \ldots, n - 1$. Our main result in Theorem 1 shows that, for the appropriate choice of $m$ and $\mathrm{n_v}$, this estimate is a BNE with an additive error $\epsilon \in (0, 1)$. Here we present the detailed proof of this theorem.

**Proof:**  The proof of the theorem comprises of two parts. The first is related to the error due to the stochastic trace estimator. The random state vector in our algorithm $|v_l\rangle = |h_{c(l)}\rangle$ is some random Hadamard column with $c(l)$ defining the random index, then the estimate $\langle h_{c(l)} | A | h_{c(l)} \rangle$ can be viewed as a uniform random sample of the transformed matrix $M = HAH^T$ with the Hadamard matrix $H$, i.e., $\langle h_{c(l)} | A | h_{c(l)} \rangle = \langle e_{c(l)} | M | e_{c(l)} \rangle$ where $|e_l\rangle$ are basis vectors. Hence, we can use the analysis of unit vector estimators in (Avron & Toledo, 2011) to obtain error bounds. In particular, we apply Theorem 16 of (Avron & Toledo, 2011).

**Lemma 1.** *(Avron & Toledo, 2011, Theorem 16) Assume we are given a Hermitian matrix $A \in \mathbb{R}^{N \times N}$, error tolerance $\epsilon \in (0, 1)$ and probability parameter $\eta \in (0, 1)$. Then, for random state vectors $|v_l\rangle = |h_{c(l)}\rangle$ as random Hadamard columns, $l = 1, \ldots, \mathrm{n_v}$, and for $\mathrm{n_v} \geq \frac{r_H^2(A) \log(2/\eta)}{\epsilon^2}$ where $r_H(A) = \max_i A_{ii}$, we have*

$$\mathbb{P}\left( \left| \frac{1}{\mathrm{n_v}} \sum_{l=1}^{\mathrm{n_v}} \langle v_l | A | v_l \rangle - \mathrm{trace}(A) \right| \leq \epsilon \cdot N \right) \geq 1 - \eta . \tag{10}$$

The proof follows from the arguments establishing Theorem 16 in (Avron & Toledo, 2011), where we set $t = \epsilon \cdot N$ in the Hoeffding's inequality, the samples take values in the interval $[0, \max_i M_{ii}]$, and we know $M_{ii} = N \cdot A_{ii}$ since $H$ has orthogonal columns with $\|h_i\|^2 = N$.

The second part is due to the error in the Chebyshev polynomial approximation of the step function. For our analysis, since the step function is a discontinuous function, we consider a surrogate

function to approximate using the Chebyshev polynomials. In particular, we consider the following polynomial, which was considered in (Musco & Musco, 2015) for the analysis of Krylov subspace methods.

**Lemma 2** (Chebyshev Minimizing Polynomial (Musco & Musco, 2015)). *Let $\alpha > 0$ be a specified parameter, and the gap $\gamma \in (0, 1]$, and let $q \geq 1$. Then, there exists a degree $m$ polynomial $p(x)$ such that:*

- $p((1 + \gamma)\alpha) = (1 + \gamma)\alpha,$
- $p(x) \geq x$ *for all* $x \geq (1 + \gamma)\alpha,$
- $|p(x)| \leq \frac{\alpha}{2^{m\sqrt{\gamma}-1}}$ *for all* $x \in [0, \alpha].$

*Furthermore, when $q$ is odd, the polynomial only contains odd powered monomials and the polynomial is given by*

$$p(x) = (1 + \gamma)\alpha \frac{T_m(x/\alpha)}{T_m(1 + \gamma)}.$$

Here $T_m(x)$ is the $m$-degree Chebyshev polynomial of the first kind. Utilizing this polynomial, we have the following result.

**Proposition 1.** *The Betti number estimate $\xi$ given by*

$$\xi = \frac{\text{trace}(\tilde{p}(\tilde{\Delta}_k))}{|S_k|}, \tag{11}$$

*where $\tilde{p}(x) = p(1 - x)$ and $p(\cdot)$ is the polynomial in Lemma 2, for parameters $\alpha = (1 - \delta)$, and $\gamma = \frac{\delta}{1-\delta}$ and a degree $m \geq \frac{\log(1/\epsilon)}{\sqrt{\delta}}$, satisfies*

$$\left| \xi - \frac{\beta_k}{|S_k|} \right| \leq \epsilon.$$

*Proof.* Suppose the eigenvalues of $\tilde{\Delta}_k$ are in the interval $\{0\} \cup [\delta, 1]$, then the eigenvalues of $I - \tilde{\Delta}_k$ will be in the interval $[0, 1 - \delta] \cup \{1\}$, and by Lemma 2 with $\alpha = (1 - \delta)$ and $\gamma = \frac{\delta}{1-\delta}$, we have

$$\beta_k - |S_k| \frac{\alpha}{2^{m\sqrt{\gamma}-1}} \leq \text{trace}(\tilde{p}(\tilde{\Delta}_k)) \leq \beta_k + |S_k| \frac{\alpha}{2^{m\sqrt{\gamma}-1}},$$

since $(1 + \gamma)\alpha = 1$ and the function is $|\tilde{p}(x)| \leq \frac{\alpha}{2^{m\sqrt{\gamma}-1}}$ for all $x \in [\delta, 1]$. For selecting an appropriate degree $m$, we want $\frac{\alpha}{2^{m\sqrt{\gamma}-1}} \leq \epsilon$, for which $m \geq \frac{\log(\alpha/\epsilon)}{\sqrt{\gamma}}$ suffices. By substituting the values we conclude that

$$m \geq \frac{\log(1/\epsilon)}{\sqrt{\delta}}$$

suffices. □

We are now ready to complete the proof of the main theorem. The stochastic Chebyshev method approximates the trace as $\text{trace}_{n_v}(\tilde{p}(\tilde{\Delta}_k)) = \frac{1}{n_v} \sum_l \langle v_l | \tilde{p}(\tilde{\Delta}_k) | v_l \rangle$, for random vector states $|v_l\rangle$. The Betti number estimate $\chi_k$ can then be written as $\chi_k = \frac{\text{trace}_{n_v}(\tilde{p}(\tilde{\Delta}_k))}{|S_k|}$. We therefore need to bound

$$\left| \chi_k - \frac{\beta_k}{|S_k|} \right| = \frac{1}{|S_k|} \left| \text{trace}_{n_v}(\tilde{p}(\tilde{\Delta}_k)) - \beta_k \right|.$$

By triangle inequality, we have

$$\left| \text{trace}_{n_v}(\tilde{p}(\tilde{\Delta}_k)) - \beta_k \right| \leq \left| \text{trace}_{n_v}(\tilde{p}(\tilde{\Delta}_k)) - \text{trace}(\tilde{p}(\tilde{\Delta}_k)) \right| + \left| \text{trace}(\tilde{p}(\tilde{\Delta}_k)) - \beta_k \right|.$$

From Lemma 1, since the maximum diagonal entry of $\tilde{\Delta}_k$ is 1, the size is $|S_k|$ and $|\tilde{p}(x)| \leq 1$ for $x \in [0, 1]$, and we obtain for $n_v = O(\frac{\log(2/\eta)}{\epsilon^2})$

$$\left| \text{trace}_{n_v}(\tilde{p}(\tilde{\Delta}_k)) - \text{trace}(\tilde{p}(\tilde{\Delta}_k)) \right| \leq \epsilon \cdot |S_k|.$$

Moreover, from Proposition 1, we have

$$\left| \text{trace}(\tilde{p}(\tilde{\Delta}_k)) - \beta_k \right| \leq \epsilon \cdot |S_k|.$$

**Additional errors:** In addition to these errors, during the actual hardware implementation, we will encounter additional errors due to noise. Two sources of noise exist, namely (a) shot noise due to measurement and (b) hardware noise. The shot noise is typically modelled using a Gaussian assumption, and hence is assumed to reduce as $O(1/\sqrt{T})$ for $T$ repeated measurements/shots. This implies that the Chebyshev moments we compute will have errors. Suppose the additive error/noise in the moment computations is $\epsilon_T$ after $T$ shots and the noise is independent. That is, each moment we compute $\theta_l^{(j)} = \langle v_l| T_j(\tilde{\Delta}_k)|v_l\rangle \pm \epsilon_T$ for $j = 0, \ldots, m$. Then, the error due to shot noise in the normalized Betti number estimation $\chi_k = 1 - \frac{1}{n_v}\sum_{l=1}^{n_v}\left[\sum_{j=0}^m c_j\theta_l^{(j)}\right]$, under the independent noise assumption, will be

$$\text{err}_{\text{shot}} = \sum_{j=0}^m c_j\epsilon_T.$$

If we use the step function expansion, then we have $|c_j| \leq (2/j)$ and the shot noise error will be $\text{err}_{\text{shot}} \leq 4\epsilon_T$. If we consider the polynomial in the analysis above, then we only have the $m$th degree polynomial, and $|c_m| \leq 1/|T_m(1+\gamma)| \leq 1$. The shot noise error will be even lower. For the expansion of any analytic function (e.g., we can consider a scaled $\tanh$ function for rank estimation (Ubaru et al., 2021)), the Chebyshev coefficients decay exponentially (Trefethen, 2019).

The hardware noise is much more difficult to characterize, since it is hardware and technology dependent. Therefore, in order to account for errors due to these two sources of noise, we will need to repeat the whole experiment several times and draw statistics to compute the Betti numbers.

Combining the results yields the desired bound in the theorem.

## C.2  COMPLEXITY ANALYSIS

We now discuss the circuit and computational complexities of our proposed algorithm and show that it is NISQ implementable under certain conditions, such as clique-dense complexes which commonly occur for large resolution scale and high order $k$. The main quantum component of the algorithm comprises the computation of $\theta_l^{(j)} = \langle v_l|T_j(\tilde{\Delta}_k)|v_l\rangle$, for $j = 0, \ldots, m \sim O(\log(1/\epsilon)/\sqrt{\delta})$, with $n_v \sim O(\epsilon^{-2})$ random Hadamard vectors. The random Hadamard state preparation requires $n$ single-qubit Hadamard gates in parallel and $O(1)$ time.

For a given $k$, constructing $\tilde{\Delta}_k$ involves implementing the boundary operator $\tilde{B}$ and the projectors $P_\Gamma$ and $P_k$. The operator $B$, involving the sum of $n$ Pauli operators, can be implemented using a circuit with $O(n)$ gates. Constructing $P_k$ requires $O(n\log^2 n)$ gates, and this succeeds for a random order $k$. Then for $P_\Gamma$, we need to find all the simplices that are in the complex $\Gamma$. This can be achieved in two ways. The first is the measure-and-reset approach which uses $n/2$ auxiliary qubits in parallel and $n-1$ rounds, and thus the time complexity is $O(n)$. The number of gates required will be $O(n^2)$. The second approach is to use $\binom{n}{2}$ auxiliary qubits, one per edge, and uncompute when done. Here, since we can consider $n/2$ edges at a time, the time and depth of this approach will also be $O(n)$ (once an initial projection from uniform is successful, which takes $O\left(\frac{n}{\zeta}\right)$ time).

Since we need to construct $T_j(\tilde{\Delta}_k)$ up to the power $m = O(\log(1/\epsilon)/\sqrt{\delta})$, the circuit has a total gate complexity of $O(n^2\log(1/\epsilon)/\sqrt{\delta})$ with a depth of $O(n\log(1/\epsilon)/\sqrt{\delta})$. The first projection $P_k$ yields a random order $k$, and the subsequent projections onto a simplicial order needed for higher moments will also have to be onto the same order $k$. Due to the application of the boundary operator $B$, the subsequent simplicial order projections will result in a projection onto one of the simplicial orders $k-1$ or $k+1$ (after one application) and $k-2$, $k$ or $k+2$ (after two applications). Hence, we need to repeatedly apply the order projection (a constant number of times) in order to ensure that we are operating on the right order (in addition to the complex projection). The procedure of computing the $m$ Chebyshev moments is repeated $n_v = O(\epsilon^{-2})$ times with different random Hadamard column vectors, and thus the total time complexity of our algorithm to compute the BNE $\chi_k$ is given by

$$O\left(\frac{1}{\epsilon^2}\max\left\{\frac{n\log(1/\epsilon)}{\sqrt{\delta}}, \frac{n}{\zeta}\right\}\right).$$

That is, we need $O\left(\frac{n}{\zeta}\right)$ time for the initial projection $P_\Gamma$ to succeed, and then $O(n\log(1/\epsilon)/\sqrt{\delta})$ for Chebyshev moments estimation using qubitization. Suppose $\delta_k$ is the spectral gap of $\Delta_k$ and $\tilde{\Delta}_k = \frac{\Delta_k}{n}$, then $\delta = \frac{\delta_k}{n}$.

## C.3  QUANTUM ADVANTAGE

Table 1 summarizes the circuit and computational complexities of our algorithm and compares them to that for the QTDA algorithm of Lloyd et al. (2016). As remarked earlier, the gate and time complexities for this QTDA algorithm reported in Gyurik et al. (2020) and Gunn & Kornerup (2019) are different from those reported in Lloyd et al. (2016), since Gyurik et al. (2020) and Gunn & Kornerup (2019) both assume the operator $\tilde{B}$ is given and thus they add the complexities of the two steps (Grover's algorithm and QPE); refer to Remark 1 above.

Table 1: Comparisons of the circuit and computational complexities for QTDA to compute BNE with an $\epsilon$ error, a $\zeta$ fraction of order-$k$ simplices in the complex, and a $\delta$ smallest nonzero eigenvalue of $\tilde{\Delta}_k$.

| Methods | # Qubits | # Gates | Depth | Time |
|---|---|---|---|---|
| Lloyd et al. (2016) | $2n + \log n + \frac{1}{\delta}$ | $O\left(\frac{n^2}{\delta\sqrt{\zeta}}\right)$ | $O\left(\frac{n^2}{\delta\sqrt{\zeta}}\right)$ | $O\left(\frac{n^4}{\epsilon^2\delta\sqrt{\zeta}}\right)$ |
| Ours (NISQ-QTDA-1) | $3n/2$ | $O(n^2\log(1/\epsilon)/\sqrt{\delta})$ | $O(n\log(1/\epsilon)/\sqrt{\delta})$ | $O\left(\frac{1}{\epsilon^2}\max\left\{\frac{n\log(1/\epsilon)}{\sqrt{\delta}}, \frac{n}{\zeta}\right\} \times \|c\|_2^2\right)$ |
| Ours (NISQ-QTDA-2) | $\tilde{O}(n^2)$ | $O(n^2\log(1/\epsilon)/\sqrt{\delta})$ | $O(n\log(1/\epsilon)/\sqrt{\delta})$ | $O\left(\frac{1}{\epsilon^2}\max\left\{\frac{n\log(1/\epsilon)}{\sqrt{\delta}}, \frac{n}{\zeta}\right\}\right)$ |

In NISQ-TDA-1, auxiliary qubits in $P_\Gamma$ are heavily reused due to the power of measurement and reset. Unfortunately, this qubit-saving strength comes at a large sampling cost factor which can only be ignored for constant $\delta$ [7] and $\|c\|_2^2$, which is the 2-norm of the coefficients of the Chebyshev approximation to the step function. NISQ-TDA-2 uses reversible computation and qubitization, completely avoiding the large sampling overhead, but instead requiring $\binom{n}{2}$ auxiliary qubits.

**Simplices/Clique dense complexes:** We first discuss examples of complexes that are simplices/clique dense. Gyurik et al. (2020) presented a few examples of a family of graphs that are clique-dense. Using the clique-density theorem (Reiher, 2016), we can consider a class of graphs/complexes that are clique-dense. Let $\gamma > \frac{k-2}{2(k-1)}$ be a constant. Then, for a graph with $n$ nodes and $\gamma n^2$ edges and for a given order $k \geq 3$, we have $|S_k| = \Omega(n^{k+1})$ by the clique-density theorem (Reiher, 2016). If $\gamma \geq \frac{k-1}{k}$, then the graph will be even denser. Such clique-dense complexes occur in TDA when the resolution scale $\varepsilon$ is large (close to maximum distance between points), and therefore QTDA algorithms can achieve a significant speedup for BNE over classical algorithms, particularly when we are interested in larger (and many) orders of $k$. We also refer to the discussions in Lloyd et al. (2016); Gyurik et al. (2020) on when quantum TDA algorithms are advantageous.

**Laplacian spectral gap:** We next discuss different settings, namely when the Laplacian of a given simplicial complex has a sufficiently large spectral gap such that a small degree $m$ will suffice for BNE. Not much is known for general simplicial complexes in terms of lower bounds for $\delta$, the smallest nonzero eigenvalue of the combinatorial Laplacians (Gyurik et al., 2020). However, we can identify many specific examples of simplicial complexes for which $\delta$ can be large. Indeed, several articles (Goldberg, 2002; Horak & Jost, 2013; Yamada, 2019; Lew, 2020a;b) have studied the spectra of the Laplacian of different simplicial complexes, including random simplicial complexes (Gundert & Wagner, 2016; Kahle, 2016; Knowles & Rosenthal, 2017; Adhikari et al., 2020; Beit-Aharon & Meshulam, 2020).

*Some specific complexes:* First, let us consider a few specific types of simplicial complexes. The articles by Horak & Jost (2013) and Yamada (2019) consider the Laplacian spectra of $k$-regular

---

[7] Details of this corrected runtime (see acknowledgements and the section on Qubitization), an alternate approach and improvements to the qubitization approach is in preparation, Akhalwaya et al.

complexes and orientable complexes. A simplicial complex $\Gamma$ is $k$-regular if and only if all of its $k$-faces have the same degree $d_k$, whereas a $k+1$-dimensional simplicial complex $\Gamma$ is *orientable* if and only if all $k$-faces of $\Gamma$ have orientation such that any two simplices which intersect on a $(k-1)$-face induce a different orientation on that face. For $k$-regular simplicial complexes with degree $d_k = 1$, the Laplacian $\Delta_k$ has all nonzero eigenvalues equal to $k + 2$. Horak & Jost (2013) show similar results for higher degree and for orientable $k$-dimensional simplicial complexes. Yamada (2019) presents lower bounds on the nonzero eigenvalues of the Laplacian for these two types of complexes in terms of the Ricci curvature (Bauer et al., 2011) of the complex. For an orientable $k$-dimensional simplicial complex $\Gamma$ with maximum degree $d_k$ for the $(k-1)$-faces, the smallest nonzero eigenvalues of $\Delta_k$, denoted by $\delta_k$, satisfies

$$\delta_k \geq (k+1)(\kappa_c - 1) + \frac{2}{d_k},$$

where $\kappa_c$ is the Ricci curvature on $\Gamma$. If the complex is orientable $k$-regular, then the minimal eigenvalues of $\Delta_k$ satisfies $\delta_k \geq (k+1)\kappa_c$. We refer to Yamada (2019) for bounds on the Ricci curvature $\kappa_c$ for $k$-regular complexes. Such complexes therefore can have a large spectral gap between zero and nonzero eigenvalues (i.e., large $\delta$ for the scaled Laplacian) when $k$ is sufficiently large.

Next, the article by Goldberg (2002) considers the Laplacian spectra of a few specific complexes. For a finite simplicial complex $\Gamma$ that contains distinct flapoid clusters of size $d_c$, the nonzero eigenvalues of the Laplacian are all equal to $d_c = o(n)$. The article by Lew (2020b) presents a lower bound for the spectral gap of the $k$-Laplacian $\Delta_k$ for complexes without missing faces. In particular, for an $n$-vertex simplicial complex $\Gamma$ without missing faces of dimension larger than $\ell$, the smallest nonzero eigenvalue (spectral gap) of $\Delta_k$, for $k \leq \ell$, satisfies

$$\delta_k \geq (\ell+1)(d_k + k + 1) - \ell n,$$

where $d_k$ is the minimal degree of a $k$-simplex in $\Gamma$. These complexes therefore can also have a large spectral gap, under appropriate conditions.

*Random complexes:* Let us now consider random simplicial complexes. For a random complex $\Gamma$ with $n$ vertices and constants $C_1, C_2$ and $p \geq (k + C_1) \log(n)/n$, Gundert & Wagner (2016) show that, if the expected degree of $k - 1$ faces is $d_k := p(n - k)$, then the normalized Laplacian[12] $\hat{\Delta}_k$ has all its nonzero eigenvalues in the interval

$$\left[1 - \frac{C_2}{\sqrt{d_k}}, 1 + \frac{C_2}{\sqrt{d_k}}\right],$$

with high probability. It was recently shown by Adhikari et al. (2020) that, for random dense graphs/complexes, the limiting spectral gap (between zero and nonzero eigenvalues) of the normalized Laplacian approaches $1/2$. Another interesting and relevant result related to the spectra of random complexes was obtained by Beit-Aharon & Meshulam (2020), who consider random subset complexes. Suppose $\Gamma$ is a full complex (also called a homological sphere) with all possible simplices of order up to $n - 1$, i.e., an $n$-simplex. Let $\tilde{G}$ be a random subset of $\Gamma$, $\tilde{G} \subset \Gamma$, of size $\tilde{n}$ and let $\delta_k$ be the minimal (smallest nonzero) eigenvalue of the $k$-Laplacian $\Delta_k$ of $\tilde{G}$ for $k < n$. Then, for $k \geq 1$ and $\xi > 0$, if the size $\tilde{n} = \lceil \frac{4k^2 \log n}{\xi^2} \rceil$, we have (Beit-Aharon & Meshulam, 2020)

$$\mathbb{P}\left[\delta_k < (1 - \xi)\tilde{n}\right] \leq O\left(\frac{1}{n}\right).$$

These results therefore suggest that random dense complexes will likely have a large spectral gap between zero and nonzero eigenvalues. Indeed, this is exactly the regime (large $\zeta$) where the quantum algorithms are advantageous. As discussed by Lloyd (1996), such dense complexes occur in TDA when the resolution scale $\varepsilon$ is large. For such complexes, our proposed QTDA algorithm has great prospects to be NISQ implementable.

**Approximate BNE:** When the spectral gap of the Laplacian $\tilde{\Delta}_k$ is not larger than the chosen threshold $\delta$, our NISQ-QTDA algorithm estimates an approximate Betti number by counting the

---

[12]The normalized Laplacian is defined as $\hat{\Delta}_k := D_k^{-1} \Delta_k$, where $D_k$ is the diagonal matrix with the degrees of the faces as its diagonal entries.

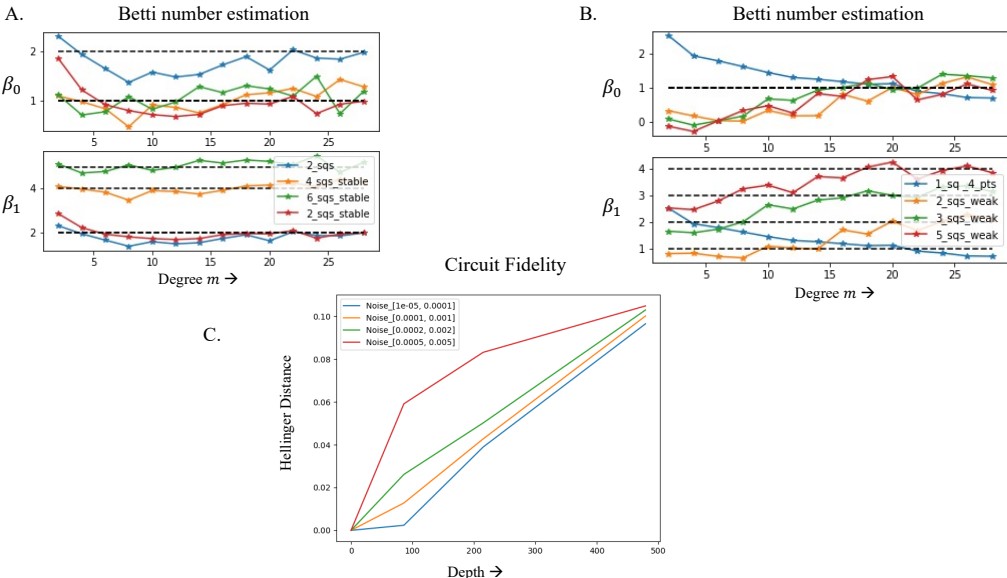

Figure 7: Results from classical and noisy simulations: Betti number estimation as a function of Chebyshev degree $m$ for (A.) simplicial complexes with stable connections/shape, i.e., large spectral-gap and (B.) complexes with weak connections/shape. C. Fidelity of NISQ-QTDA circuit (measured using Hellinger distance) as a function of circuit depth for different noise-levels.

(larger) eigenvalues above the threshold $\delta$. This was defined in Gyurik et al. (2020) as the problem of approximate Betti number estimation (ABNE). Such ABNE will be useful in certain situations, since our method provides an approach to filter out small (noisy) eigenvalues and only consider larger (dominant) eigenvalues for estimating the Betti number. These small nonzero eigenvalues occur when there are thinly (loosely) connected components in the complex. Such connections likely occur when the resolution scale $\varepsilon$ is small, and they might not persist when $\varepsilon$ increases. Our approach therefore provides a way to filter out noise and estimate the features that persist at larger resolution scale. Moreover, we note that computing the moments of the Laplacians $\Delta_k$ (exponential in size) for different $k$ is non-trivial, and these moments can be used as features for certain downstream learning tasks, for example.

# D  ADDITIONAL EXPERIMENT RESULTS

In this section, we present a few additional experimental results that provide further insights into TDA and our NISQ-QTDA.

## D.1  CLASSICAL TDA RESULTS

We first illustrate the performance of the classical version of the proposed stochastic Chebyshev method for Betti number estimation. Figure 7 (A and B) plot the Betti number estimated as a function of the Chebyshev polynomial degree $m$ for two classes of simplicial complexes, respectively. The first class in Figure 7(A) correspond to complexes that have stable shape (well connected), as in if we remove few edges, the shape (and hence the Betti numbers) do not change. The Laplacian corresponding to such complexes have a large spectral gap. We note that the Betti number estimated by our method for such complexes are fairly accurate and a small degree polynomial approximation suffices to get a good estimate of the Betti numbers. The black dash lines are the true Betti numbers and the star-solid lines correspond to the Betti numbers estimated for four different complexes all with 8 vertices, respectively (2 squares, 4 squares, 6 squares/cube and 2 squares with diagonals included).

The second class in Figure 7(B) correspond to complexes that have weak connections, such that removing a few edges changes the shapes (and Betti numbers) significantly. Such complexes have relatively Laplacian small spectral gap. We note that, we require a higher degree approximation for accurate Betti number estimation. We considered four different complexes (1 square and 4 dangling points, 2 squares with two edges connecting them, 3 squares, and 5 squares). Interestingly, we observe that the Betti vs. degree curves for the 3 similar complexes have a similar shape and we can be distinguish between them, even though the Betti numbers estimated are not correct. This shows our method is provides a way to distinguish complexes/shape even with a low degree approximation.

Figure 7(C) plots the Hellinger distance (a measure of fidelity) of our NISQ-QTDA circuit as a function of the circuit depth, i.e., number of vertices/qubits for different noise levels in simulations. The square Hellinger distance between two probability measures $P$ and $Q$ is defined as

$$H^2(P, Q) = \frac{1}{2} \int \left( \sqrt{dP} - \sqrt{dQ} \right)^2.$$

In the figure, we consider the Hellinger distance between the measured probabilities using noiseless simulations and noisy simulations. Four different noise levels were considered. The noise-level pairs show the 1-qubit and 2-qubit gate error rates. The measurement error rate was set to be same as the 2-qubit gate error rate. We plot the Hellinger distance restricted to the top $10\%$ of the noise-free outcomes, thereby focusing only those outcomes with sufficient probability mass at the given shot count. As expected, the Fidelity/noise-error increases as the circuit depth (and number of qubits) increase.

## D.2 CMB RESULTS

The cosmic microwave background (CMB) is the remnant after-glow of the Big Bang, forming an opaque background curtain in the sky. The curtain dates back to when the Universe was about $380,000$ years old (redshift $z \approx 1100$) when the universe first became transparent to radiation. CMB photons produce a nearly perfect black-body spectrum with a present day temperature of around $2.725$ K (Aghanim et al., 2020). This temperature fluctuates by around one part in $10^4$ depending upon the angle in the sky one looks. The uniformity corroborates an epoch of cosmological inflation and provides a window into physics in the early Universe. Recalling that a function $\phi$ is Gaussian when the vector $\vec{v} = (\phi(x_1), \ldots, \phi(x_n))$ is drawn from an $n$-dimensional Gaussian distribution for all $n$ and for all $x_i$, of particular interest are deviations from Gaussianity in the CMB. These are typically assessed by looking at three-point and four-point correlation functions, the spatial bispectrum and trispectrum. An application of TDA compares simulations of the CMB with a Gaussian probability distribution to those with particular local injections of non-Gaussianity, *e.g.*, by considering $\varphi(x) = \phi(x) + f_{\mathrm{NL}}\left(\phi(x) - \langle\phi\rangle\right)^2$, with $\phi(x)$ a Gaussian field and $f_{\mathrm{NL}}$ a parameter capturing the amount of non-Gaussianity. Preliminary investigations in this direction using classical TDA include (Cole & Shiu, 2018a; Feldbrugge et al., 2019a; Biagetti et al., 2021), and the present state of the art detects $f_{\mathrm{NL}} \sim \mathcal{O}(10)$. Benchmarking against different values of $f_{\mathrm{NL}}$, NISQ-TDA could compute higher-order Betti numbers associated with the actual sky and test for non-Gaussianity in the CMB to a potentially greater degree of sensitivity.

Here, we present preliminary results to illustrate how we might apply the insights of Adler et al. (2014) (for example drawing high-dimensional points and excluding the 'core') for the detection of non-Gaussianity in the CMB data. We consider random sets of $n = 64$ sample points in three dimensions from Gaussian and exponential distributions. The visualization of the data distributions in 3D and the norm distributions are presented in Figure 8 (first two rows). Our goal is to illustrate that by only using homological information (Betti numbers in the form of 'persistent diagrams') of very small sample sets (forced to have zero-mean and unit-variance, making the task much more difficult), we can still distinguish the two distributions. Furthermore, we wish to show that each extra Betti number considered boosts the accuracy. Such an illustration would suggest that we can detect non-Gaussianity in CMB data using NISQ-QTDA to a greater degree than low-order classical TDA since QTDA furnishes all high probability Betti numbers of any order.

In order to run our experiment, we use the GUDHI (Maria et al., 2014) package for Betti number estimation of the data at different resolutions $\varepsilon$ to create the persistence diagrams (representing occurrence/birth and disappearance/death of holes at resolution scale $\varepsilon$). See the last row of Figure 8 for an example output of the GUDHI pipeline. Directly visualizing the points and the norm dis-

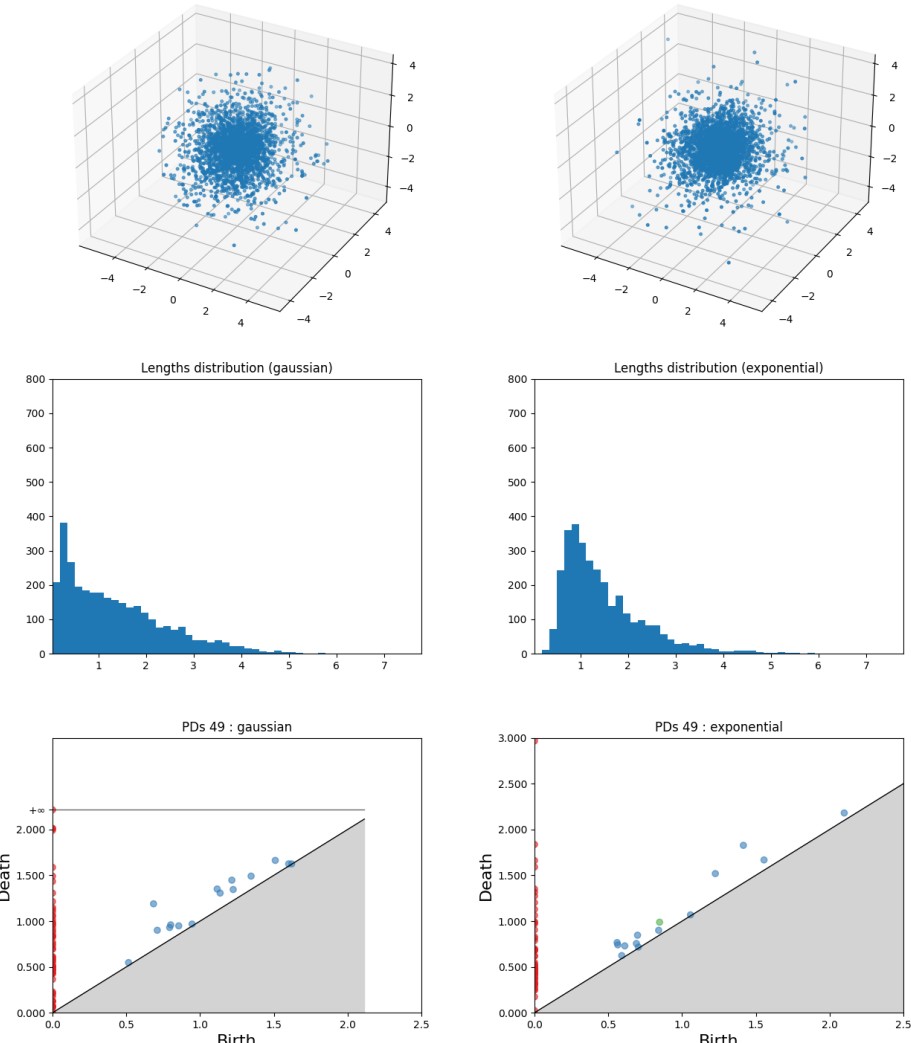

Figure 8: Results for CMB data: Random $n = 64$ sample points per sample set in three dimensions from the Gaussian distribution (in left column) and the exponential distribution (right column). The first row is the visualization of all 50 sample sets (30 train and 20 test). The second row is the histogram of norms of all the points, and the last row are example persistence diagrams for two sample sets, one from each distribution.

tributions illustrates that the distinguishability task is non-trivial (especially when the form of the distribution is unknown). Secondly, it is only when viewing the persistence diagrams (PDs) does the differing behavior become easier to detect. However, a single sample set's associated PD is unlikely to be rigorously helpful in telling apart another single sample set. Fortunately, in the simulation use-case as well as for the real CMB data (considering it's presumed rotational invariance) we may repeatedly draw sample sets, creating a set of PDs to train and test on. Therefore, we next employ a Bayesian (learning) package called BayesTDA (Maroulas et al., 2020) with train (30) and test (20) sample sets, to calculate two Poisson point-process posteriors from the training data, allowing for the calculation of the Bayesian likelihoods of the test data given the two competing posteriors, and culminating in a single Bayes factor for each test data set. The Bayes factors can then be used for classification by comparing to a threshold. To obviate the choice of a threshold we simply calculate the Area Under the Curve (AUC) measure of classification accuracy (0.5 is as bad as random guessing and 1 is perfect classification).

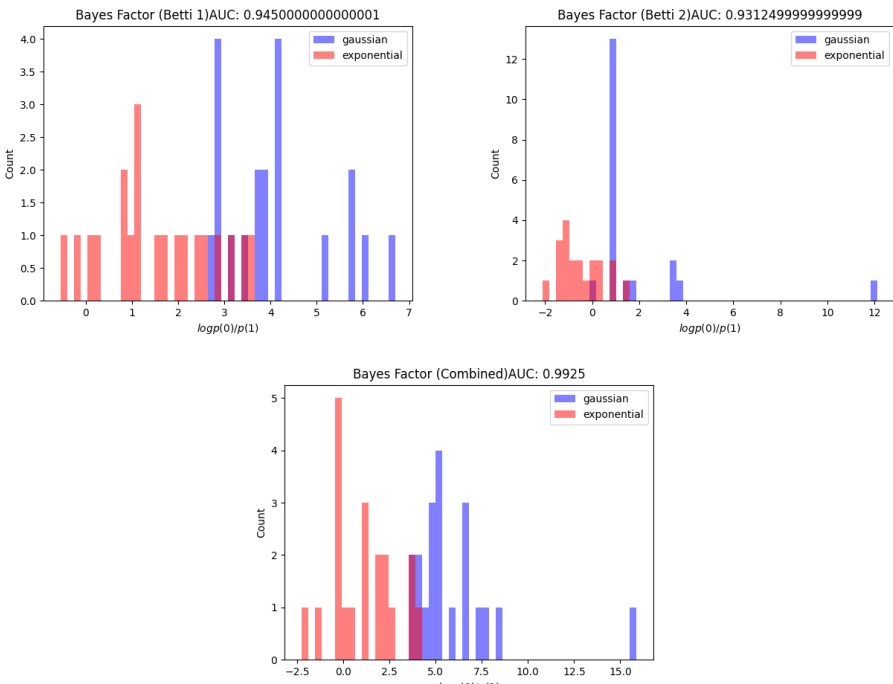

Figure 9: Results for simulated 'CMB' data (Gaussian vs exponential (Adler et al., 2014) , but with an enforced mean of zero and variance of unity (element-wise)): Bayes factors and AUC obtain for the data distributions when $\beta_1$ (top left), $\beta_2$ (top right) and both $\beta_1$ and $\beta_2$ combined (bottom) were used in the learning model

In Figure 9, we present the Bayes factors and AUC obtained for the data distributions when $\beta_1$ (top left), $\beta_2$ (top right) and both $\beta_1$ and $\beta_2$ combined (bottom) were used in the learning model. We observe that we can clearly distinguish the two distributions using the Bayes factors. Moreover, we note that the different Betti numbers contain independent valuable information. Therefore, by potentially computing all relevant Betti numbers of the CMB data using NISQ-QTDA, and using the above approach, we expect to detect non-Gaussianity in the CMB data to higher sensitivity levels than is possible classically (since even for moderate numbers of data points, we cannot compute higher order Betti numbers classically).

