# OpenReview forum: "Topological data analysis on noisy quantum computers"
_ICLR.cc/2024/Conference — ICLR 2024 oral_

### Official Review · Reviewer_FCsJ · 2023-10-23

**Soundness:** 3 good
**Presentation:** 3 good
**Contribution:** 3 good
**Rating:** 8
**Confidence:** 2

**Summary:**

The paper proposes a quantum algorithm for topological data analysis (TDA), which is a technique for extracting shape-related features of high-dimensional data. The proposed algorithm NISQ-TDA uses a quantum rejection sampling technique to project onto the data-defined simplicial complex, and a stochastic rank estimation method to estimate the Betti numbers, which are signature values that describe the shape of the data. The paper provides theoretical and empirical analyses of the algorithm, showing that it has error guarantees, short circuit depth, noise resiliency, and potential speedup over classical algorithms for certain classes of problems.

**Strengths:**

1. The writing of the paper is clear and well-structured. The paper proposes a quantum algorithm for TDA called NISQ-TDA, which is a technique for extracting shape-related features of high-dimensional data. NISQ-TDA, is designed to work on noisy intermediate-scale quantum devices, which are the current and near-term quantum computers that have limited resources and error rates.

2. The paper presents one of the first quantum machine learning algorithms with short depth and potential significant speedup under certain assumptions. The proposed algorithm neither suffers from the data-loading problem nor does it likely require fault-tolerant coherence for even mid-size datasets.

3. The paper presents results from implementing the entire algorithm on real quantum hardware and noisy simulations, illustrating noise-resiliency at realistic noise-levels. The paper also discusses possible applications of NISQ-TDA for scientific machine learning and AI tasks.

**Weaknesses:**

1. The reviewer has a basic understanding of quantum computing, but not familiar with TDA. The paper does not provide sufficient background and related work on quantum computing, TDA and QTDA. It assumes that the reader is familiar with these topics and does not cite relevant literature or explain the key concepts and notations.

2. The paper does not provide any empirical evidence of the quantum advantage or noise-resiliency of the NISQ-TDA algorithm. It only shows some preliminary results on small datasets and noisy simulations, without any statistical analysis or comparison with baselines.

**Questions:**

Please see the weaknesses.

---

> ### Author Response · Authors · 2023-11-17
>
> We thank the reviewer for their valuable comments. Please find our responses below.
>
> "The reviewer has a basic understanding of quantum computing, but not familiar with TDA. The paper does not provide sufficient background and related work on quantum computing, TDA and QTDA. It assumes that the reader is familiar with these topics and does not cite relevant literature or explain the key concepts and notations"
>
> R: We thank the reviewer for this point. Our work brings together three distinct fields (algebraic topology/homology, quantum computing, and machine learning), and includes a new algorithm, rigorous theoretical results, experimental results, and new applications. Unfortunately, it was difficult to fit all the different details within 9 pages. Therefore, we tried to include detailed descriptions for each of these in the Appendix (of about 20 additional pages), where we provide pictorial illustrations, flow charts, circuit diagrams, detailed equations, and descriptions to supplement the main paper.
> We will further revise the main paper to address this comment and point to the relevant portions of the appendix for more details.
>
> "The paper does not provide any empirical evidence of the quantum advantage or noise-resiliency of the NISQ-TDA algorithm. It only shows some preliminary results on small datasets and noisy simulations, without any statistical analysis or comparison with baselines."
>
> R: As we discuss in the paper, in order to demonstrate real quantum advantage, we will require quantum computers (QCs) with at least a few hundred qubits that are very well connected (or have extremely efficient "swap" operations) and have very high multi-qubit gate fidelity. Unfortunately, such devices are likely few years away, and we will not be able to demonstrate true quantum advantage for NISQ-TDA using existing quantum computers.
> Using current quantum computers, we can compute Betti numbers of small datasets, as demonstrated in our paper, and for such datasets, exact Betti numbers can be easily known. To the best of our knowledge, the baseline QTDA algorithm cannot be implemented on current hardware without fault tolerance, and there are no classical baselines to compute high-order Betti numbers (the problem is NP hard).
> We wish to note that real quantum advantage has not been demonstrated on any practically useful problems so far. We strongly believe our work makes an important contribution in this direction, by presenting a novel quantum algorithm for a useful problem (that is known to be classical hard), which does not suffer from data loading problems, requires short depth and with potential speedup.   We present rigorous error guarantees for the algorithm and an actual end-to-end implementation of the algorithm on a real hardware. We will revise the discussion in the paper to make all these points more clearly.

---

> > ### Comment · Reviewer_FCsJ · 2023-11-18
> > **Official Comment by Reviewer**
> >
> > Thank you for the detailed response. The reviewer hopes to see the authors provide a more easily understandable discussion in the revised paper, as this would be very meaningful for other readers who wish to comprehend and follow along. I have already raised the score.

---

### Official Review · Reviewer_tAjA · 2023-10-29

**Soundness:** 4 excellent
**Presentation:** 3 good
**Contribution:** 4 excellent
**Rating:** 8
**Confidence:** 2

**Summary:**

This paper proposes a hybrid classical-quantum algorithm for solving the problem of topological data analysis. More specifically, they consider the problem of estimating the Betti numbers for a simplicial complex. They propose an algorithm that is NISQ-friendly yet still yields speedup over the best-known classical algorithms. Moreover, the algorithm is fully implemented on an existing ion trap device, showing good agreement with simulated results as well as robustness to noise.

**Strengths:**

- This is a very interesting work that proposes to solve a problem that is hard for classical computers, accessible to quantum speedup, and is practically useful.
- The end-to-end implementation on NISQ devices is quite amazing and the close match with noiseless simulation is also surprising.
- One particularly interesting aspect of this work is how they avoid the data-loading issue which is typical for many proposed quantum algorithms for machine learning. The projections by mid-circuit measurements appears to be an important step and I wonder if other quantum algorithms can benefit from this.

**Weaknesses:**

For real quantum advantage, the input data must satisfy several conditions listed at the end of Section 3. The advantage for solving the problem of deciding whether a simplicial complex has exponentially many holes seems less clear and perhaps less practical. It would be great to know if the algorithm still provides speedup for real-world instances.

**Questions:**

- I am a bit confused by Figure 1B, C, and D. What exactly are the bars representing? I assume it shows the probabilities of obtaining results corresponding to vertices, edges, triangles, etc., but is it the case that several are omitted for the cube and square?
- I wonder if there is any intuitive, high-level reasoning for what kind of problem structure is being leveraged here that enables quantum speedup.

---

> ### Author Response · Authors · 2023-11-17
>
> We thank the reviewer for their valuable and constructive comments. Please find our response below.
>
> "For real quantum advantage, the input data must satisfy several conditions listed at the end of Section 3. The advantage for solving the problem of deciding whether a simplicial complex has exponentially many holes seems less clear and perhaps less practical. It would be great to know if the algorithm still provides speedup for real-world instances."
>
> R: This is a good point. Indeed, in order to achieve real quantum advantage (super-polynomial to exponential speedup), we require simplicial complexes with exponentially many holes, and it is not clear whether such complexes exist in real-world data. However, Schmidhuber and Lloyd (2022) recently showed that the original QTDA algorithm achieves a quadratic (to 4th power) speedup over classical algorithms. Since our algorithm improves the runtime of QTDA, we should expect a further speedup over classical algorithms for real-world instances. We will revise the paper to clarify and emphasize these points.
>
>
> Questions:
>
> "I am a bit confused by Figure 1B, C, and D. What exactly are the bars representing? I assume it shows the probabilities of obtaining results corresponding to vertices, edges, triangles, etc., but is it the case that several are omitted for the cube and square?"
>
> R: This is correct. We have plotted the fractional counts (probabilities) measured after applying the Laplacian circuit once for the datasets. We have plotted the probabilities for only those states, which have significant enough counts (more than 1% of the total counts). As expected, most of the states will have negligible probabilities/counts, particularly for larger $n$.
> We will clarify exactly what the bars are representing in Figure 1.
>
> "I wonder if there is any intuitive, high-level reasoning for what kind of problem structure is being leveraged here that enables quantum speedup.""
>
> R: At a high-level, we are taking advantage of the following fact: A simplicial complex with $n$ points can have up to $2^n$ simplices of all possible $n$ orders, each of these simplices of any given order $k$ can be represented as basis vectors/states of a certain Hilbert space $H_k$, and the union of these is simply the $n$-qubit Hilbert space. Thus, we can represent any of the $2^n$ simplices as an $n$-qubit state. Next, we take advantage of quantum parallel search (since a uniform superposition state represents existence of all simplices), and that a $k$-simplex is fully defined by the $k$ edges (small input). Therefore, entangling qubits pairwise suffices to represent the simplices present in the complex.
> We appreciate this point and will add to the paper a discussion of such intuitive, high-level reasoning (also see the Appendix for a brief discussion on this).

---

> > ### Comment · Reviewer_tAjA · 2023-11-22
> >
> > Thanks for the responses. I'll keep the score.

---

### Official Review · Reviewer_MnCf · 2023-10-29

**Soundness:** 4 excellent
**Presentation:** 3 good
**Contribution:** 4 excellent
**Rating:** 8
**Confidence:** 3

**Summary:**

This paper presents NISQ-TDA, a new quantum algorithm for topological data analysis (TDA) that is readily implemented on near-term noisy quantum devices. This work avoids an unrealistic data input model by constructing an explicit (and shallow) quantum circuit for data-loading. The two major steps in the circuit construction are (1) representing the full boundary operator as a Fermionic boundary operator that allows efficient Pauli decompositions, and (2) projection onto problem-specific simplices using a quantum rejection sampling technique. Notably, the explicit circuit construction of the combinatorial Laplacian operator does not require accessing stored quantum data.

To estimate the Betti number (defined as the rank of the kernel space of the boundary operator), the authors adopt a stochastic rank estimation method. First, the rank estimation problem is recast into a trace estimation problem through a spectral mapping function $h(\cdot)$, which can be constructed using a truncated Chebyshev series. Then, by employing a stochastic trace estimation method due to Hutchinson, the Betti number is estimated by summing over finitely many the Chebyshev moments.

NISQ-TDA has been tested on both noisy numerical simulators and trapped-ion quantum devices (Quantiuum). Numerical results suggest good robustness against machine noise.

**Strengths:**

The Quantum TDA problem was first studied by Lloyd et al. (2016), in which an efficient quantum algorithm was proposed. The algorithm in Lloyd et al. requires a fault-tolerant quantum computer to run Grover's algorithm and digital Hamiltonian simulation (for QPE). This paper appears to propose a new quantum algorithm that does not rely on a fault-tolerant quantum computer with potential superpolynomial speedups compared to classical.

Empirically, this paper presents both resource analysis (Fig 1A) and real-machine results (Fig 1B-D). Also, a noisy simulation suggests this algorithm is robust to machine noise. The numerical evidence strongly implies that this algorithm could be useful for NISQ devices, justifying the claim by the authors. Potential applications to ML, AI, neuroscience, and cosmology are discussed.

Overall, this paper is well-written and the plots are easy to follow.

**Weaknesses:**

I feel the technical discussion in Section 3 (especially the projection to a simplicial order) is a bit hard to follow. Is it possible to give some concrete examples (e.g., in terms of explicit circuit & Pauli operators)? Also, it would be better to discuss more on the actual resources (gate counts, # of measurements, etc.) spent on the Quantinuum hardware. See the following "Questions" section.

**Questions:**

1. Theorem 1 gives a rigorous sample complexity of NISQ-TDA and the total time complexity is $O(\frac{n\log(1/\epsilon)}{\sqrt{\delta} \epsilon^2 \zeta_k^{2\log(1/\epsilon)/\sqrt{\delta}}})$. This time complexity looks exponentially better than the best-known classical result. However, this result is not directly comparable with the complexity of Quantum TDA ($O(n^5/(\delta_k \sqrt{\zeta_k}))$) (of course, the QTDA requires stronger quantum computers). Is it possible that NISQ-TDA can outperform the fault-tolerant QTDA in a certain parameter regime?

2. The circuit depth of NISQ-TDA heavily depends on the Chebyshev truncation number. What is the exact dependence of the Chebyshev truncation number in terms of the input data set (or the projection operator)? Does the resource analysis (Fig 1A) treat the Chebyshev truncation number as a parameter depending on the number of vertices, or it is fixed as a constant?

3. In Fig 1B-D, how many shots were used on the hardware to estimate the probability?

4. In Fig 2A, the error seems huge for intermediate-size problems even with moderate machine noise (e.g., (0.001, 0.01) for 1- and 2-qubit gate error). To solve practical problems in the application domains, is there any way to further suppress/mitigate the machine noise for NISQ-TDA?

---

> ### Author Response · Authors · 2023-11-17
>
> We thank the reviewer for their valuable and constructive comments. Please find our responses below.
>
> "I feel the technical discussion in Section 3 (especially the projection to a simplicial order) is a bit hard to follow. Is it possible to give some concrete examples (e.g., in terms of explicit circuit & Pauli operators)? Also, it would be better to discuss more on the actual resources (gate counts, # of measurements, etc.) spent on the Quantinuum hardware."
>
> R: We thank the reviewer for this valuable suggestion. We have addressed your suggestion in the Appendix of the revised paper by adding two example circuits for the two projection operations (projection onto the complex $P_{\Gamma}$ and projection onto the order $P_k$), respectively; please see Figures 6 and 7. For the experiments on the Quantinuum hardware, for the (2, 4, 8) vertices experiments, the number of measurements were $10^3$, $10^4$, and $2\times 10^4$, respectively.
>
> Questions:
>
> "1. Theorem 1 gives a rigorous sample complexity of NISQ-TDA and the total time complexity. This time complexity looks exponentially better than the best-known classical result. However, this result is not directly comparable with the complexity of Quantum TDA (of course, the QTDA requires stronger quantum computers). Is it possible that NISQ-TDA can outperform the fault-tolerant QTDA in a certain parameter regime?"
>
> R: This is a valid point. Under NISQ or fault-tolerance regime, our NISQ-TDA algorithm should certainly require a shorter circuit depth and fewer gates, compared to the QTDA algorithm. The overall runtime (the number of times we must repeat) will then depend on the parameters, the error tolerance $\epsilon$, the spectral gap $\delta$, and the fraction of simplices $\zeta$. Our algorithm has similar order dependency on $\epsilon$, quadratically better dependency on $\delta$, but worse dependency on $\zeta$, compared to QTDA. Therefore, for dense complexes (when $\zeta \approx O(1)$), our algorithm should outperform QTDA in any quantum regime.
> We will revise the text to make these points clear.
>
>
> "2. The circuit depth of NISQ-TDA heavily depends on the Chebyshev truncation number. What is the exact dependence of the Chebyshev truncation number in terms of the input data set (or the projection operator)? Does the resource analysis (Fig 1A) treat the Chebyshev truncation number as a parameter depending on the number of vertices, or it is fixed as a constant?""
>
> R: The degree of the truncated Chebyshev polynomial $m$ will depend on the spectral gap as $1/\sqrt{\delta}$; see Theorem 1. In the Appendix, we present many simplicial complex examples and discuss the behavior of their spectral gap with respect to the number of vertices. For certain simplicial complexes, the spectral gap is constant, while for others the gap depends inversely on the number of vertices.
> We will improve the discussion of these points in the main paper and point to the appendix for more details.
>
> In Fig 1A, the plots are for fixed degrees $m=1$ and $m=3$, respectively, which we will clarify.
>
> "3. In Fig 1B-D, how many shots were used on the hardware to estimate the probability?""
>
> R: For the (2, 4, 8) vertices experiments, the number of shots used were $10^3$, $10^4$, and $2\times 10^4$, respectively, which we will clarify in the revised paper.
>
> "4. In Fig 2A, the error seems huge for intermediate-size problems even with moderate machine noise (e.g., (0.001, 0.01) for 1- and 2-qubit gate error). To solve practical problems in the application domains, is there any way to further suppress/mitigate the machine noise for NISQ-TDA?""
>
> R: This is a good question. For intermediate-size problems, NISQ-TDA will require circuit depths of a few hundred, and if the gate fidelities are not very high (moderate depolarizing noise levels in the gates), then the measurements will likely be very noisy without some form of error correction, and hence, the errors will be high. A key advantage of NISQ-TDA is that it includes multiple projection operators (involving measure and reset operations), which naturally suppress noise to an extent (as discussed in Section 3). It would certainly be interesting to further investigate whether certain error correction/noise mitigation mechanisms can be incorporated with our algorithm.
>
> We would also like to note that some of the latest quantum computers already have better noise performance (than (0.001, 0.01) noise levels for 1- and 2-qubit gate errors).

---

> > ### Comment · Reviewer_MnCf · 2023-11-22
> >
> > The response from the authors addresses my comments in the review. I'll keep my score.

---

### Official Review · Reviewer_G9wH · 2023-10-30

**Soundness:** 3 good
**Presentation:** 3 good
**Contribution:** 4 excellent
**Rating:** 8
**Confidence:** 4

**Summary:**

Finding any application of NISQ (noisy intermediate-scale quantum) technology has been challenging, let alone in machine learning problems. Furthermore, most NISQ algorithms are heuristics in nature and do not come with rigorous guarantees. The authors propose a new NISQ algorithm for topological data analysis (TDA) with a rigorous performance guarantee for solving TDA efficiently (no classical algorithms can solve TDA efficiently due to complexity-theoretic hardness), does not suffer from data loading problems (many previous quantum algorithms only obtain advantage when one neglects the cost in data loading), and is robust to noise.

**Strengths:**

Understanding potential applications of NISQ is a very important question in the entire field of quantum computing. The work provides a significant step toward obtaining an end-to-end application of NISQ by proposing a NISQ algorithm for topological data analysis.

Topological data analysis is only efficient classically for low-order Betti numbers. Calculating high-order Betti numbers is known to be hard on a classical computer assuming widely believed complexity-theoretic conjectures. Hence, there are quantum advantages in calculating high-order Betti numbers.

While quantum algorithms for TDA have been known and have been subject to extensive studies in the past few years, existing quantum algorithms require deep quantum circuits, making them challenging to run on NISQ computers. The work proposes an algorithm NISQ-TDA that uses significantly shallower quantum circuits, making the algorithm suitable for the current quantum technology.

The authors experimentally tested the proposed NISQ-TDA algorithm on a 12-qubit trapped-ion quantum computer and showed promising results that the proposed NISQ algorithm is robust to realistic device noise.

**Weaknesses:**

A minor weakness of this work is that the proposed algorithm is not strictly applicable to NISQ devices. NISQ devices can only implement an O(log n)-depth quantum circuit before the measurement outcomes become random noise. While the work provided a significant improvement in the circuit depth, the circuit depth is still O(n).

A theorem analyzing the amount of local depolarizing noise on each gate that can be tolerated by the proposed NISQ-TDA algorithm is missing in the current writeup (the physical experiments do show that the proposed algorithm is promising).

**Questions:**

Could the authors analyze the amount of local depolarizing noise on each gate that NISQ-TDA can tolerate? I suspect that the algorithm cannot tolerate a constant amount of noise per qubit (which is how people typically think of NISQ). However, understanding the noise level can still be very useful (e.g., noisy random quantum circuit sampling requires 1/n noise to have exponential quantum advantage).

---

> ### Author Response · Authors · 2023-11-17
>
> We thank the reviewer for their valuable and constructive comments. Please find our responses below.
>
> "A minor weakness of this work is that the proposed algorithm is not strictly applicable to NISQ devices. NISQ devices can only implement an O(log n)-depth quantum circuit before the measurement outcomes become random noise. While the work provided a significant improvement in the circuit depth, the circuit depth is still O(n)."
>
> Response: This is a fair point that (current) NISQ devices will likely yield reasonable results only for sub-linear depth circuits, particularly for large '$n$'.  However, there has been much progress recently, with IBM demonstrating interesting results with 100-qubits 100-depth circuits, and Quantinuum's latest H2 model (32-fully-connected qubits) having high (one and two qubit) gate fidelities that can obtain reasonable results with deep circuits (circuit depths of few hunderds). Hence, we certainly hope that in the near future, NISQ devices will be capable of achieving good results (maintain coherence) with linear depth circuits. We will revise the text to make these points clear.
>
>
> "A theorem analyzing the amount of local depolarizing noise on each gate that can be tolerated by the proposed NISQ-TDA algorithm is missing in the current writeup (the physical experiments do show that the proposed algorithm is promising).
>
> Could the authors analyze the amount of local depolarizing noise on each gate that NISQ-TDA can tolerate? I suspect that the algorithm cannot tolerate a constant amount of noise per qubit (which is how people typically think of NISQ). However, understanding the noise level can still be very useful (e.g., noisy random quantum circuit sampling requires 1/n noise to have exponential quantum advantage)."
>
> Response: This is a very good point, and our simulation results indeed demonstrate this point. In our noisy simulation experiments (Figure 2), we study the performance of the algorithm under different 1-qubit and 2-qubits gate depolarizing noise levels, along with measurement noise. We used the Qiskit noise simulator (qiskit.providers.aer.noise) for our experiments. The noise levels depicted in Figure 2 represent the depolarization probabilities that we used for the 1-qubit and 2-qubits gate depolarizing errors, respectively (using the depolarizing_error function in Qiskit). The ReadOut error probabilities were set to be the same as the 2-qubits gate depolarization probabilities, consistent with what we can expect in today's real hardware.
> In our experiments, we considered different orders of depolarizing noise levels, but they were fixed with respect to the number of qubits '$n$'. As the reviewer rightly suspected, for a fixed noise level, we observe that the overall error grows at least polynomially with the number of qubits. We will improve our discussion of these results to address this important point raised by the reviewer.

---

> > ### Comment · Reviewer_G9wH · 2023-11-21
> >
> > The response from the authors addresses the minor questions I have.
> > I think this is a good paper so I would keep the score of 8.

---

### Meta-Review · Area_Chair_QRRq · 2023-12-04

**Metareview:**

This paper studies the problem of topological data analysis, which is widely applied for extracting complex and valuable shape-related summaries of high-dimensional data. This task is intractable in general, and the current submission extends to the study of quantum computing algorithms for topological data analysis. In particular, this paper presents NISQ-TDA, a fully implemented end-to-end quantum machine learning algorithm needing only a short circuit-depth, that is applicable to high-dimensional classical data, and with provable asymptotic speedup for certain classes of problems. The algorithm was also executed on real-world quantum machines, and empirical results also suggest that the algorithm is robust to noise.

This paper has notable strengths:
- It studies quantum algorithms of a machine learning problem in an end-to-end sense, and a rigorous quantum speedup in proved.
- The algorithm was executed on real-world quantum machines (Quantinuum H1)
- In addition, various numerical experiments were conducted, including resource analysis noisy simulation Potential applications to different subjects are discussed, including neural networks, cosmic microwave background, neuroscience, and genetics.

There are some minor weaknesses of the paper, such as Section 3 is a bit technical and hard to follow, the circuit depth of O(n) might still be an obstacle for current NISQ devices, and the concepts of TDA can be better explained. The rebuttals have adequately discussed these points and made further clarifications.

In the final version, the authors should merge the rebuttal into relevant parts of the paper.

**Justification For Why Not Higher Score:**

N/A

**Justification For Why Not Lower Score:**

This paper is a great work demonstrating the potential of executing machine learning algorithms on quantum computers. It's impressive that this work made significant contributions on both theory (provable guarantee of quantum speedup by the proposed quantum algorithm, NISQ-TDA) and experiments (the algorithm can already be performed on current real machines with positive results, and there are also numerical experiments justifying its robustness). The wide range of potential applications (neural networks, cosmic microwave background, neuroscience, and genetics) also shapes the importance of this submission.

I think this result is worth being accepted as an oral to highlight recent advances in the interdisciplinary topic between machine learning and quantum computing. In particular, its theory, real machine experiments, and potential applications as a whole piece can be very interesting for general audiences of ICLR 2024 to learn.

---

### Decision · Program_Chairs · 2024-01-16

Accept (oral)